# Propagation from meteorological to hydrological drought in the Horn of Africa using both standardised and threshold-based indices

Rhoda A. Odongo [1], Hans De Moel [1], and Anne F. Van Loon [1]

[1]Institute of Environmental Studies, Vrije Universiteit Amsterdam, Netherlands

*Correspondence to*: Rhoda A. Odongo (rhodaachieng.odongo@vu.nl)

## Abstract

There have been numerous drought propagation studies in data-rich countries, but not much has been done for data-poor regions (such as the Horn of Africa (HOA)). In this study, we characterise meteorological, soil moisture and hydrological droughts and the propagation from one to the other for 318 catchments in the HOA to improve understanding of the spatial variability of the drought hazard. We calculate the Standardised Precipitation Index (SPI), Standardised Soil Moisture Index (SSMI) and Standardised Streamflow Index (SSI). In addition, we use the variable threshold method to calculate the duration of drought below a predefined percentile threshold for precipitation, soil moisture and discharge. The relationship between meteorological and soil moisture is investigated by finding the SPI accumulation period that has the highest correlation between SPI and SSMI and the relationship between meteorological and hydrological drought is analysed by the SPI accumulation period that has the highest correlation between SPI and SSI time series. Additionally, we calculated these relationships with the ratio between the threshold-based meteorological drought duration and soil moisture drought duration, and the relation between threshold-based meteorological drought duration and streamflow drought duration. Finally, we investigate the influence of climate and catchment characteristics on these propagation metrics. The results show that (1) the propagation from SPI to SSMI and the mean drought duration ratio of meteorological to soil moisture (P/SM) are mainly influenced by soil properties and vegetation, with the short accumulation periods (1 to 4 months) of SPI in catchments with arable land, high mean annual precipitation, and low sand and silt content, while longer accumulations (5 to 7 months) are in catchments with low mean annual upstream precipitation and shrub vegetation; (2) the propagation from SPI to SSI and precipitation to streamflow duration ratio are highly influenced by the climate and catchment control, i.e., geology, elevation and landcover, with the short accumulation times in catchments with high annual precipitation, volcanic permeable geology, and cropland, and the longer accumulations in catchments with low annual precipitation, sedimentary rocks and shrubland; and (3) the influence of mean annual upstream precipitation is more important for the propagation from SPI to SSI than from SPI to SSMI. Additionally, precipitation accumulation periods of approximately 1 to 4 months in wet western areas of HOA, and of approximately 5 to 7 months in the more dryland regions are found. This can guide forecasting and management efforts as different drought metrics are thus of importance in different regions.

## 1   Introduction

The Horn of Africa (HOA) experiences recurrent droughts (including the current multi-year drought), which have severe impacts such as crop losses, livestock deaths and diseases, as well as frequent emergencies, food insecurity, infrastructure damage and high economic costs (IGAD & WFP, 2017). This is particularly devastating for smallholder farmers whose livelihoods depend on rain-fed agricultural systems and livestock (IGAD & WFP, 2017).

Over the past decade, studies have been conducted in the HOA to understand and characterise extreme events such as droughts. Most of these studies use modelled data due to the lack of observational data in the region. The lack of observational data, especially on river discharge, has led to limited analysis of hydrological drought events. Several of the studies have focused on meteorological and agricultural drought rather than hydrological droughts (Agutu et al., 2017, 2020; Anderson et al., 2012; Awange et al., 2016; Belal et al., 2014; Dutra et al., 2013; Edossa et al., 2010; Gebrechorkos et al., 2020; Haile et al., 2019, 2020; Kurnik et al., 2011; Lyon, 2014; Nicholson, 2014; Rulinda et al., 2012; Tonini et al., 2012). In these studies, drought was assessed based on soil moisture (model and reanalysis), precipitation (satellite-based, observed and a combination of both), terrestrial water storage (TWS; through the Gravity Recovery and Climate Experiment) and normalised difference vegetation index (NDVI).

Soil moisture and hydrological drought, which have a strong impact on agriculture and water use in ecosystems and society respectively, have devastating effects in the HOA (Shukla and Wood, 2008; Van Loon, 2015). Therefore, it is critical for water resource management to understand how the drought signal transitions from aberrant meteorological conditions to soil moisture and eventually to hydrological drought. This process is called propagation. Drought

propagation is strongly influenced by climate and catchment characteristics (Barker et al., 2016; Van Loon and Laaha, 2015; Van Loon and Van Lanen, 2012). Therefore, the combined effects of climate and catchment characteristics on propagation of droughts also need to be assessed to better understand the underlying processes of drought development. There are many studies on drought propagation studies in data-rich countries (e.g. USA, China), but not much has been done for the data-poor regions (e.g. HOA). In this study, we define drought as a prolonged period of below-average water availability. Drought is usually classified into three types: meteorological (precipitation deficit), agricultural or soil moisture (soil moisture deficit), and hydrological (abnormally low water levels in rivers, reservoirs, lakes and groundwater) (He et al., 2013; Huang et al., 2017; Jiang et al., 2019; Van Loon et al., 2016).

Drought frequency, severity and duration are important characteristics of drought events and can be used to study drought propagation. Many studies have quantified these drought characteristics using standardised indices (i.e., Standardised Precipitation Index (SPI) (Mckee et al., 1993), Standardised Soil Moisture Index (SSMI) (Hao and Aghakouchak, 2014) and Standardised Streamflow Index (SSI) (Huang et al., 2017)). Some studies have also used threshold-based indices to calculate the duration and deficit of drought (as a measure of severity) and to investigate drought propagation (Heudorfer and Stahl, 2017; Tallaksen et al., 2009; Van Lanen et al., 2013; Van Loon, 2013; Van Loon et al., 2014; Van Loon and Laaha, 2015). Most of these studies are at the catchment level (Apurv et al., 2017; Huang et al., 2017; Tallaksen et al., 2009) and some at the regional level (Barker et al., 2016; Van Loon, 2013; Van Loon and Laaha, 2015; Xu et al., 2019a). These studies mainly focused on the drought characteristics, identifying some characteristics related to lagging, attenuation, lengthening and pooling. What remains unclear in these studies is how the propagation of drought from meteorological to soil moisture is related to climate and catchment characteristics. Furthermore, the two approaches to drought characterisation differ in that the standardised indices do not provide information on drought deficit volumes but can be used across different geographical regions, unlike the variable threshold-based method which preserves the hydrological values but cannot be used across different geographical regions. These methods thus provide different information when used for spatial analysis of drought propagation.

Many studies have used statistical methods to assess the drought propagation and relate them to climate and catchment characteristics. Some of the studies provide an indication of which variables should be included in an analysis of drought propagation in the HOA, such as geology, landcover, mean annual precipitation and seasonal characteristics. For example, Barker et al. (2016) have characterised meteorological and hydrological droughts and their propagation in the UK. The relationship between meteorological and hydrological drought was assessed by cross correlating the 1-month SSI (SSI-1) with different SPI accumulation periods using a Pearson correlation coefficient. They also investigated the influence of climate and catchment characteristics on hydrological drought characteristics and its propagation using Pearson correlation along with Spearman correlation. They found that the SPI accumulation periods correlated differently with the SSI-1 depending on the region in the UK, which could be due to the differences in hydrogeology and mean annual precipitation. Huang et al. (2017) used SPI and SSI to characterise meteorological and hydrological droughts, respectively. They investigated the propagation time and the influence of El Nino Southern Oscillation (ENSO), Artic Oscillation (AO) and underlying surface properties on drought propagation in the Wei River Basin in China using the cross-wavelet analysis. They found that ENSO and AO are strongly correlated with actual evaporation and thus, influenced the propagation time from meteorological to hydrological drought (which is influenced by seasonal characteristics). Van Loon and Laaha (2014) used variable threshold level methods to characterise meteorological and hydrological drought in 44 Austrian catchments free of major disturbances. They analysed the combined influence of climate and catchment characteristics of drought propagation using various statistical tools (i.e., bivariate correlation analysis, regression analysis). The results showed that hydrological drought duration is primarily influenced by storage and release (i.e., base flow index, geology, and land use). In addition, the duration of meteorological drought is important for hydrological drought duration, and the hydrological drought deficit is governed by catchment wetness (mean annual precipitation).

These results cannot be easily generalised and applied to the HOA because of the different climate and hydrogeology. Thus, there is still a need for a deeper understanding of the drought propagation in this region and for appropriate indicators to characterise drought. Choosing an appropriate drought characterisation indicator is key to understanding the relationship between drought hazard and drought impacts. Depending on the area of application, different indices may prove useful (Vicente-Serrano & Lopez-Moreno, 2005). The objective of this study is therefor to understand (1) how drought characteristics change as drought propagates from meteorological to soil moisture and to hydrological drought in the HOA, and (2) which climatic and catchment characteristics influence the propagation from meteorological to soil moisture to hydrological drought, using both standardised and threshold-based indices. The

study is conducted in a number of catchments across Kenya, Somalia and Ethiopia with diverse hydro-climatic and geological characteristics.

## 2   Case study area

The study was conducted on a selection of catchments based on the boundaries of HydroBASINS level 6, which covers Kenya, Somalia and Ethiopia and consists of 338 catchments (Lehner & Grill, 2013). The region has a seasonal climatological regime. It is predominantly semi-arid, but ranges from very humid in the Ethiopian highlands and Mount Kenya region to very dry in parts of Somalia, southern and south-eastern Ethiopia and north-eastern Kenya. The region is mostly shrubland with cropland and forests in the very humid Ethiopian highlands, around Lake Victoria

and on the high slopes of Mount Kenya (Fig. 1a). The mean annual precipitation decreases from west to east and from high altitudes (the Ethiopian highlands and the region around Mount Kenya receive a mean annual precipitation of more than 500mm) to low altitudes (north-eastern Kenya and Somalia receive mean annual of less than 200mm) (Fig. 1b and c). Precipitation increases from the Somalia coast towards the Kenyan coast (Fig. 1c), which is also reflected in the forest cover (Fig. 1a).

The prolonged rains mostly occur from March to May (MAM), while the short rains occur from October to December (OND) due to the migration of the inter-tropical convergence zone (ITCZ) from south to north and vice versa (Awange et al., 2016). This is particularly true for Kenya and Somalia, while Ethiopia experiences a single rainy season in June to September (JJAS). The region also has a very diverse geology, ranging from rich volcanic soils on the high slopes of Mount Kenya and in the Ethiopian highlands to sedimentary rocks in the semi-arid areas of southern and south-

eastern Ethiopia, Somalia, and eastern and north-eastern Kenya (Fig. 1d). The diverse topography, climate seasonality and large number of catchments and relevant catchment characteristics make it a suitable region to study.

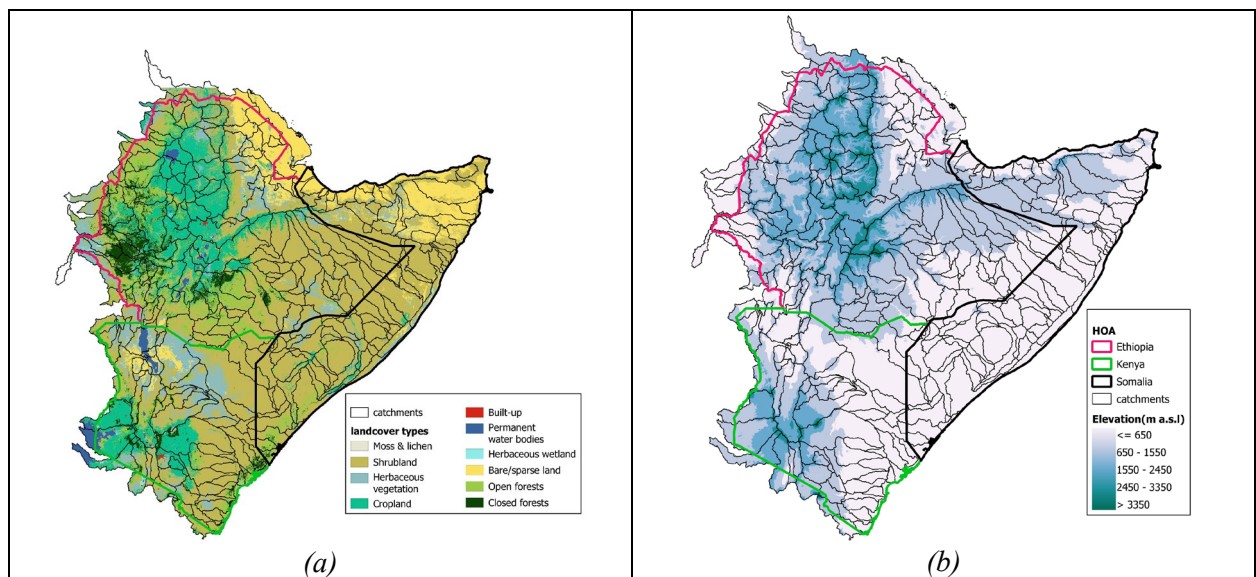

*(a)*                                                                 *(b)*

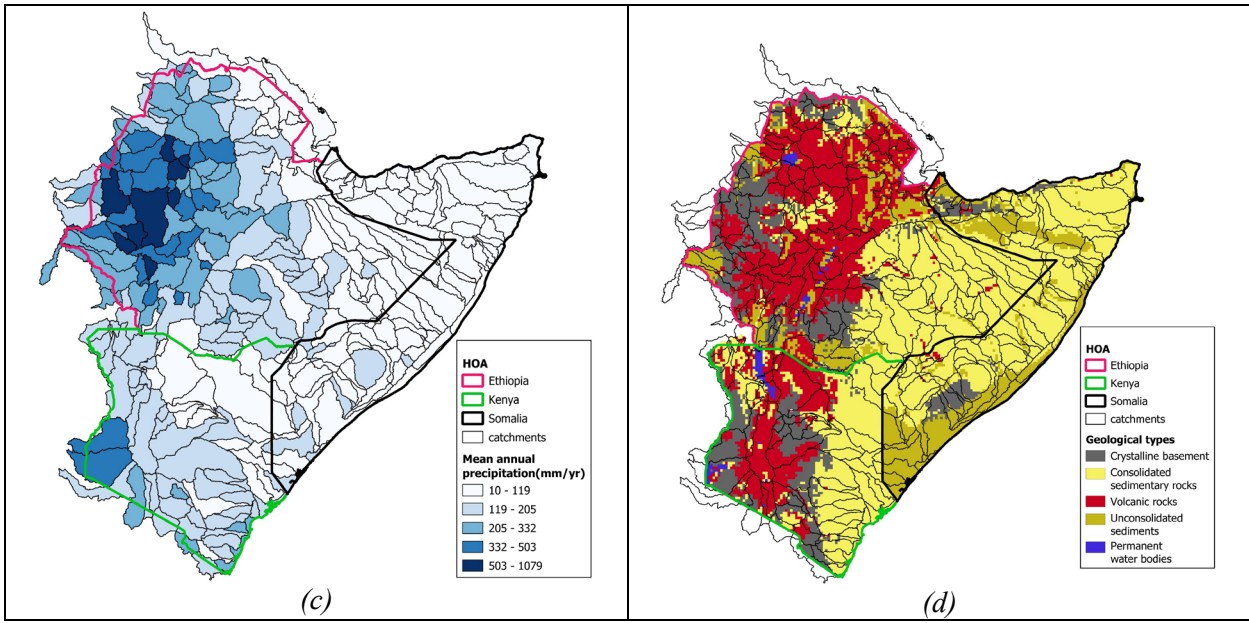

*Figure 1: Maps showing some characteristics of the study area: (a) landcover (Copernicus) (Buchhorn et al., 2020); (b) elevation (STRM) (Farr and Kobrick, 2000); (c) mean annual precipitation (MSWEP) (Beck et al., 2017b)); and (d) geology (Africa Groundwater Atlas, 2022).*

## 3   Methodology

The methodology of this study is summarised in Fig. 2. First, the data and their respective sources together with the catchment characteristics are discussed (Section 3.1), followed by calculation of the Standardised Precipitation Index (SPI), Standardised Soil Moisture Index (SSMI), and Standardised Streamflow Index (SSI) and the threshold-based indices (Precipitation to Soil moisture mean duration ratio (P/SM ratio) and Precipitation to Streamflow mean duration ratio (P/Q ratio) (Section 3.2). Then, the drought propagation analysis process is discussed (Section 3.3). Finally, the statistical analysis involving linking of both the standardised and threshold-based indices with the climate and catchment characteristics, and the comparison of both methods in characterising drought propagation is discussed (Section 3.4).

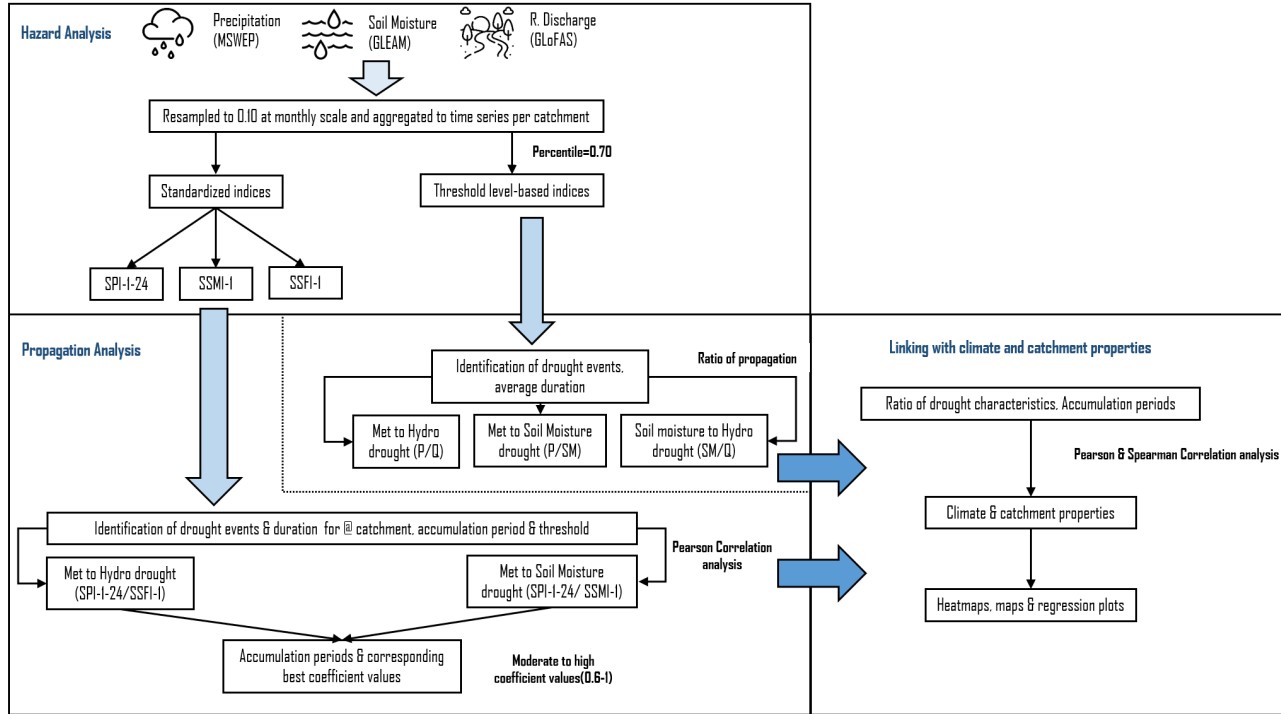

*Figure 2: Detailed methodology framework showing steps taken from hazard analysis to propagation analysis and finally to linking with climate and catchment characteristics.*

## 3.1 Data

In this study we rely on the same types of datasets used in previous studies. Ideally, propagation analysis should be based only on observed data, but this is not always feasible for large-scale analyses. In this study, we have therefore favoured data sources that are as close as possible to observed data, while still covering the entire regional range of the HOA. Accordingly, we chose to use reanalysis data for precipitation and soil moisture, but still used modelled data for streamflow because no suitable observational dataset in the HOA was found. These gridded datasets were aggregated to catchment resolution based on hydrological data and level 6 maps of HydroSHEDS (Lehner and Grill, 2013), to delineate catchments.

HydroSHEDS is a global hydrological dataset that provides information on the water drainage systems. It is based on digital elevation models (DEMs) and other geospatial data sources and is divided into several levels of detail, with level 1 being the coarsest and level 12 the finest. At each level, the dataset provides information on the location and characteristics of water bodies such as rivers, lakes and wetlands, as well as the topography of the surrounding terrain. Level 6 was chosen because it provides an average level of detail on the water drainage systems. In particular, hydrographic units (HUs) with an average size of about 10,000 square kilometres are delineated at level 6.

For the analysis, we use the upstream contributing area of each catchment. Catchments with an area of 150 square kilometres or more were selected for analysis, reducing the number to catchments to 320. Two additional catchments are excluded due to missing values (missing values were due to the resolution of soil moisture and streamflow datasets) after aggregation at the catchment level, leaving only 318 catchments. The remaining catchments provide good spatial coverage of the HOA and its diverse characteristics.

### 3.1.1 Hydrometeorological and soil moisture datasets

Precipitation data were retrieved from the Multi-Source Weighted-Ensemble Precipitation (MSWEP) version 2 (Beck et al., 2019). This is a global gridded precipitation (P) dataset covering the period 1979 to the present. MSWEP has a temporal resolution of 3 hours, daily, and monthly; and a spatial resolution of 0.1 degrees. In this study we sued the daily MSWEP precipitation data. It does not contain pure observations, but a combination of gauge-, satellite- and reanalysis-based P estimates, depending on timescale and location (Beck et al., 2017b, 2019). This dataset was selected

for this analysis due to its spatial and temporal resolution and good performance in capturing spatial and temporal variation of drought conditions (Xu et al., 2019b).

The soil moisture data were retrieved from the Global Land Evaporation Amsterdam Model (GLEAM) (version 3.5a). The model applies a set of algorithms to estimate land surface evaporation (also known as evapotranspiration) and root-zone soil moisture from satellite and reanalysis data at the global scale with a spatial resolution of 0.25 degree and a daily temporal resolution (Martens et al., 2017; Miralles et al., 2011). It uses the latest version of MSWEP precipitation (version 2.8)(Beck et al., 2017b, 2019), satellite observed soil moisture, reanalysis air temperature and radiation and vegetation optical depth (VOD)(Liu et al., 2011) to produce terrestrial evaporation and root-zone soil moisture. The root-zone soil moisture is based on the weighted average of the soil surface up to 5 centimetres (top layer), which is more variable, and the root-zone up to a layer of 100 centimetres. The GLEAM model applies the Priestley and Taylor (PT) equation (Priestley and Taylor, 1972) to calculate the Potential Evapotranspiration (PET) based on observations of European Centre for Medium Range Weather Forecasts (ECMWF), ERA-Interim surface net radiation and near surface air temperature (Dee et al., 2011). GLEAM datasets have been used in recent studies, including in the HOA (Javadinejad et al., 2019; Nicolai-Shaw et al., 2017; Peng et al., 2020). For this study, the GLEAM potential evaporation (PET) and root-zone soil moisture (see http://www.gleam.eu) were used for the period 1981–2020.

Streamflow data were retrieved from the Global Flood Awareness System (GloFAS), which consists of global gridded river streamflow data with a horizontal resolution of 0.1 degrees at a daily time step and a period from 1979 to the present (Harrigan et al., 2020). It combines both the surface and sub-surface runoff from the HTESSEL land surface model used in ECMWF's global atmospheric reanalysis (ERA5) (Balsamo et al., 2009; Hersbach et al., 2020) with the LISFLOOD hydrological and channel routing model (Hirpa et al., 2018). LISFLOOD calculates a water balance with a temporal resolution of six hours or day and a spatial resolution of 0.05 degrees (see http://www.globalfloods.eu/). The GloFAS dataset was selected because there is no observed river discharge data with sufficient spatial coverage and time period in the study region. Unfortunately, GloFAS uses ERA5 Land as precipitation input, which has been found to be less reliable in the HOA region than MSWEP or CHIRPS. Therefore, we tested the GloFAS dataset at the available discharge stations (with discharge values from 1981 onwards; total of 26 stations) in the HOA for bias compared to observed data. We found that while there is a bias in the absolute values, the anomalies are similar between the two datasets *(see for more explanation in section 5.1.1)*. Since our analysis focuses on relative deviations from normal, we deemed it acceptable to use the GloFAS data to represent discharge anomalies.

Catchment characteristics were obtained from a variety of sources. These sources include BasinATLAS (Linke et al., 2019), upstream mean annual precipitation from MSWEP (Beck et al., 2017), geological types from Africa Groundwater Atlas (Africa Groundwater Atlas, 2022) and landcover types from Copernicus Global Land Cover layers-Collection 2 (Buchhorn et al., 2020). Catchment characteristics used in this study include soil properties (i.e. percent silt, sand and clay, percent mean annual soil water content), geological types, landcover types, terrain slope, elevation, upstream contributing area, climate zones, mean annual upstream precipitation , global average aridity index (*see Table S1 Supplementary Material*). These catchment characteristics were selected because they have been found in previous studies to influence drought propagation in other regions (Barker et al., 2016; Van Loon, 2013; Van Loon and Laaha, 2015). The characteristics were also chosen because drought intensity varies according to the topographic location and the time it takes for water to flow through the catchments.

## 3.2   Drought Analysis

### 3.2.1   Standardised indices

The SPI developed by McKee et al. (1993) allows quantification of precipitation deficits or surpluses over a range of different accumulation periods. In this study, we prefer SPI over other meteorological drought monitoring indices because organizations providing climate services to the Horn of Africa, such as the IGAD Climate Prediction and Application Centre (ICPAC) use SPI specifically for drought monitoring in its East Africa Drought Watch. Several studies in the Horn of Africa have also used SPI (Kalisa et al., 2020; Okal et al., 2020; Dinku et al., 2007; Viste et al., 2013). To represent agricultural drought we selected the Standardised Soil Moisture Index (SSMI) index, and for hydrological drought we selected the Standardised Streamflow Index (SSI).

By the nature of the different indices, different distributions are best suited to fit the different data types. We used the distributions suggested by Stagge et al., (2015) for calculation of SPI, distributions suggested by Ryu and Famiglietti, (2005) for calculation of SSMI and distributions suggested by Vicente-Serrano et al., (2012) for calculation of SSI. We fitted a different distribution for each catchment, which is not a problem in our study because we analyse drought propagation with catchments and do not compare drought characteristics between catchments (*see for more explanation section 2 in Supplementary Material*). The SPI was calculated by summing daily MSWEP precipitation to obtain a monthly temporal resolution. Monthly precipitation values were fitted to a distribution, performed for each catchment, to calculate SPI values for accumulation periods ranging from 1 to 24 months. Each catchment within the HOA has a specific distribution that was either normal, gamma, exponential Weibull or lognormal for SPI calculation (Stagge et al., 2015). The number of zeros in precipitation was considered according to the recommendations of Stagge et al. (2015). In calculation of the SSMI we used mean monthly GLEAM root-zone soil moisture content and fitted normal, beta, Pearson3 or Fisk distributions (Ryu and Famiglietti, 2005). In the calculation of the SSI, we used mean monthly GloFAS streamflow values and fitted Exponential Weibull, Lognormal, Pearson3 or generalized extreme distributions (Vicente-Serrano et al., 2012). The distribution for each catchment and variable was selected based on the Kolmogorov best-fit method. Each of these distributions has been shown to fit various indices in previous studies.

All drought indices were calculated with a monthly resolution for the period 1980–2020. The standardised wet and dry periods of each indicator were included in the analysis to characterise changes in anomalies when moving through the hydrological cycle. As such, with this method, we did not define drought events, but aim to identify the anomalies over different accumulation periods.

### 3.2.2   Threshold level-based indices

The threshold-based approach is a widely used method for drought analysis (Heudorfer & Stahl, 2017; Tallaksen et al., 2009; Van Lanen et al., 2013; Van Loon, 2013; Van Loon et al., 2014; Van Loon & Laaha, 2015). In applying this approach, a drought event was defined as any event that falls below the pre-defined threshold. Drought events were identified from the monthly time series of the above hydrometeorological datasets (precipitation (P), soil moisture (SM) and river discharge (Q)) using a monthly varying threshold-based approach (without pooling) i.e. an approach that has a different value for each month (this is similar to standardised indices that fit a distribution for each month separately) to account for seasonality, and defined in terms of the duration of drought. This approach has been used in numerous studies (e.g. Beyene et al., 2014; Nyabeze, 2004; Van Huijgevoort et al., 2012; Van Huijgevoort, 2014; Van Loon, 2013; Van Loon et al., 2014; Van Loon & Laaha, 2015; Vidal et al., 2010). The 70th percentile was used as threshold. This means that each month of the year has a different threshold based on the 70th percentile of the values of the hydrometeorological variable in that month, for all years in the time series. Previous studies have used percentile ranges between the 70th and 90th (Heudorfer & Stahl, 2017; Van Loon, 2013; Van Loon et al., 2014; Van Loon & Laaha, 2015). After testing different percentile values (70th, 80th and 90th percentile), we selected the 70th percentile because it could clearly capture both moderate and severe droughts. The other percentiles were eliminated because, with the high precipitation variability experienced in the region, they had too few droughts, showed a misidentification of less severe droughts, and they did not account for most of the major known drought years. This made it difficult to identify patterns and trends (*see section 2 Fig. S5, S6 and S7 Supplementary Material*).

The duration of the drought event was determined by the total number of consecutive months in which the value of the variable was below the threshold. Then, the average duration of drought per catchment in the study area was calculated. Finally, the duration ratios were calculated (meteorological drought duration in relation to soil moisture drought duration (P/SM ratio) and streamflow drought duration (P/Q ratio)). A ratio closer to 1 indicates that the durations are similar (i.e. there is not so much clustering of meteorological droughts into streamflow droughts), while a ratio closer to 0 means there are many more meteorological droughts than streamflow droughts, indicating that they have propagated and clustered into fewer and longer P/Q droughts.

## 3.3   Drought propagation

SSMI and SSI integrate processes at the land surface and hydrogeological processes in the catchment, respectively. Therefore, comparing SSMI and SSI with SPI provides an indication of the time it takes for the drought signal to propagate through the hydrological cycle from precipitation deficits to soil moisture deficits and finally to streamflow deficits. SPI time series with accumulation periods of 1-24 months were cross-correlated against 1-month SSMI (SSMI-1) and SSI (SSI-1) time series using Pearson correlation per catchment. This cross-correlation method has been

used in many similar studies (Barker et al., 2016; Huang et al., 2017; Xu et al., 2019a) and can effectively show the similarity between different drought types. The accumulation period with the highest correlation coefficient with either SSMI-1 or SSI-1 was denoted as SPI-$n$ and used as an indication of the propagation of the meteorological drought signal to soil moisture and streamflow respectively. Only correlation values greater than or equal to 0.5 were used for the analysis of propagation, as these were considered strong signals.

Propagation times were considered short if the SPI-$n$ accumulation time was less than four months. We did not investigate whether there was a lag between the SPI and SSI time series, as other studies have found that the strongest correlations usually occur at a lag of zero months (i.e. no lag) (Barker et al., 2016). Finally, the catchments were grouped based on the calculated accumulation periods. To test the independence of the data between the catchments based on the different groups of accumulation periods, a one-tailed $t$-test was performed to see how much the groups were statistically different from each other.

In threshold-based indices, the drought propagation was studied by the ratio of the drought duration of the hydrometeorological variables. A ratio between the duration of meteorological drought and the duration of soil moisture drought (P/SM) was calculated to indicate propagation from meteorological to soil moisture drought. P/SM mean duration ratio represents the speed with which precipitation deficits affect soil moisture availability, and therefore, how quickly the ability of plants to access water is hampered during drought. A low ratio suggests that soil moisture is more resilient to precipitation deficits (slow soil moisture response to precipitation), which is probably related to catchment properties like soil type. A high ratio indicates that precipitation deficits have an faster impact on soil moisture availability (faster soil moisture response to precipitation). We also calculated the ratio of the duration of meteorological drought and streamflow drought (P/Q) to show the propagation from meteorological to streamflow drought. P/Q mean duration ratio represents the degree to which precipitation deficits affect streamflow. A low ratio suggests that streamflow is more resilient to precipitation deficits, and meteorological droughts are buffered. A high ratio indicates that precipitation deficits have a quick response in streamflow. Also these P/Q ratios are probably influenced by catchment characteristics like subsurface storage. Overall, we favoured the use of the duration ratios to other conventional indices because these ratios can provide insight into the mechanisms through which drought propagates and the vulnerabilities of different systems to precipitation deficits (Van Loon et al., 2016). These ratios take into account the effect of precipitation deficits on soil moisture and streamflow.

### 3.4 Influence of possible governing factors of climate and catchment characteristics

The effects of climate and catchment characteristics on propagation were investigated through statistical analysis. First, we analysed the strength of the relationships using cross-correlation analysis. We calculated the correlation matrix of the pairwise combinations of all variables based on Pearson correlation coefficients. Since the relationships might not be linear, we also calculated Spearman correlation coefficients and visually inspected the correlation matrix presented as a heatmap to verify the results. We created a clustered heatmap of the Pearson correlation matrix to examine the intercorrelations of the catchment characteristics. We used the Euclidean distance method to rank the coefficients. The Euclidean distance method orders rows and columns by similarity, making it easier to find groups of climate and catchment characteristics that have a joint effect on drought propagation. We then plotted individual graphs for each of the key variables against the propagation indices. In addition, we used raster zonal statistics in QGIS to link variables such as geology, landcover, climate zones, upstream areas and elevation to the indices. Second, we conducted a one-tailed $t$-test for the standardised indices of the clustered catchments. The significance test was used to determine whether the clusters of catchments differed per accumulation period.

## 4 Results and Discussion

In the following sections, drought propagation and the link with catchment characteristics per propagation indicator are discussed in detail. First, the propagation from meteorological to soil moisture drought is presented, followed by the propagation from soil moisture to streamflow drought.

## 4.1  Precipitation to soil moisture

### 4.1.1  SPI to SSMI

Mapping of SPI-*n* for SPI to SSMI propagation (Fig. 3) showed high correlation values in all catchments, especially in the south of HOA (Kenya region, average 0.82) (Fig. 3a). The high correlation values were found across the range of SPI accumulation periods. This could be due to the strong link between precipitation and soil moisture as GLEAM uses MSWEP precipitation as one of its inputs. The catchments were equally divided into short and long accumulation periods (1-4 months and 5-9 months with 159 catchments each (Fig. 3b)). The longest accumulation periods (9 and 8 months) were in the northwest of the HOA, with correlation values greater than 0.7 (Fig. 3a). Figure 3b shows that the SPI-*n* of the catchments on the north-eastern coast of HOA were between 1 and 3 months, while those of catchments at the eastern centre were longer (between 5 and 7 months). (*See section 3.1 Fig. S8, S9 and S10 Supplementary Material for spatial plots of the propagation.*)

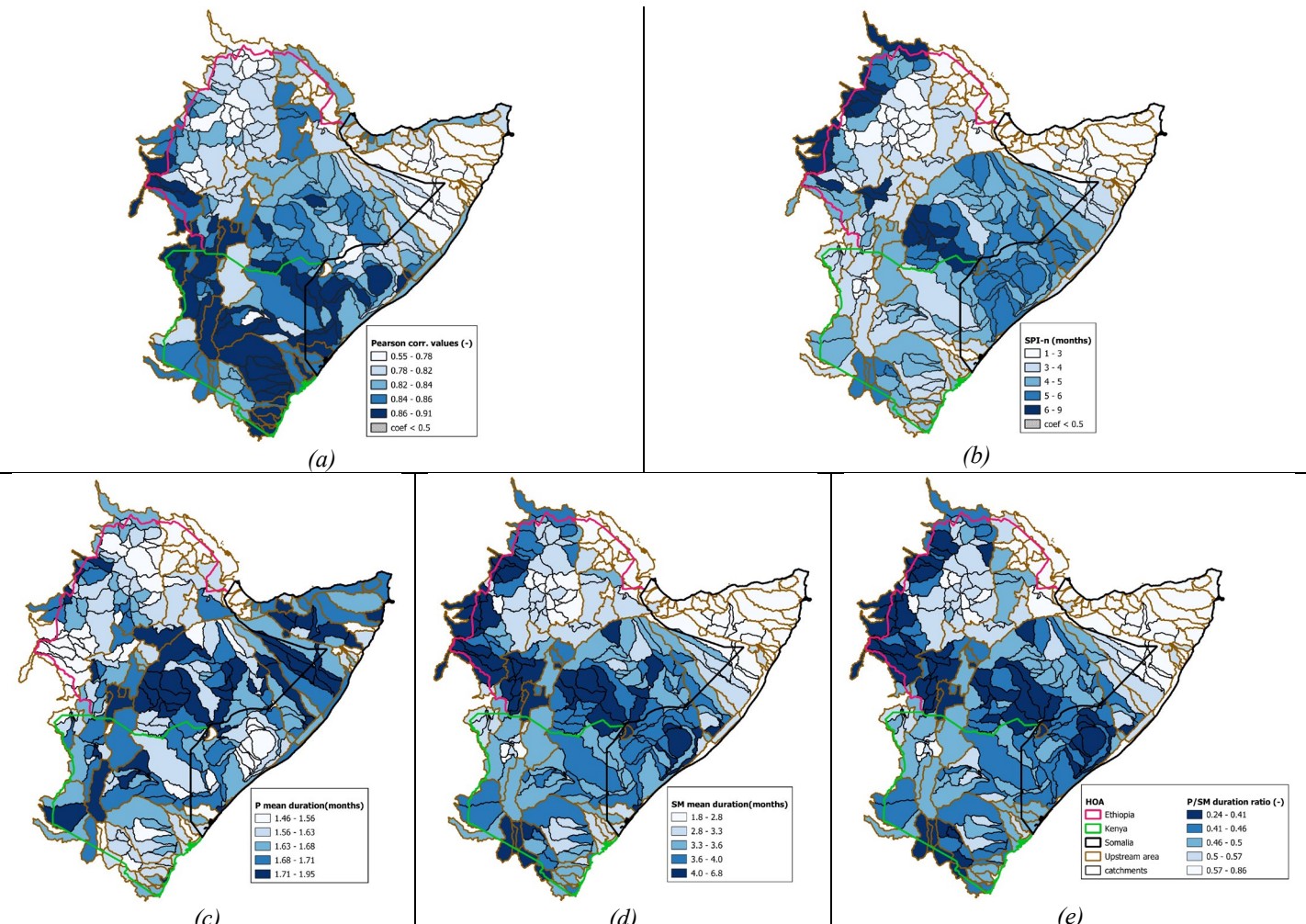

*Figure 3: Propagation from precipitation to soil moisture: (a) highest coefficient values per catchment (>0.5) from SPI to SSMI; (b) corresponding SPI-n (SPI accumulation period having highest correlation with SSMI) per catchment; (c) mean meteorological drought duration; (d) mean soil moisture drought duration; (e) ratio of drought mean duration from meteorological to soil moisture (P/SM).*

### 4.1.2  P/SM duration ratio

The duration of droughts increases as the drought signal propagates through the hydrological cycle (Fig. 3c and d). The duration of soil moisture droughts is longer than that of meteorological droughts, indicating propagation and pooling of meteorological droughts into soil moisture drought. The map of the threshold-based drought duration ratio (P/SM) (Fig. 3e) shows similar processes as the map of the propagation using standardised indices (Fig. 3b), i.e. a

short precipitation to soil moisture response in the northeast (represented by short accumulation periods in the standardised indices and a high mean duration ratio in the threshold indices (P/SM ratio)).

The analysis of the P/SM ratio (Fig. 3e) shows that the north-western centre and north-eastern centre of the HOA have high ratios, which means that soil moisture in this area responds faster to precipitation (less pooling of meteorological drought events). The P/SM ratio decreases towards the southeast coast of the HOA (in some catchments the ratio is as

low as 0.3), indicating longer soil moisture droughts towards the southeast coast of the HOA. The ratios are also low in the catchments at the north-western tip and west of the HOA, indicating longer soil moisture droughts and shorter meteorological droughts (Fig. 3c and d), implying that soil moisture in these catchments responds more slowly to precipitation (greater clustering of meteorological droughts).

### 4.1.3    Relation of precipitation-to-soil moisture with climate and catchment
characteristics

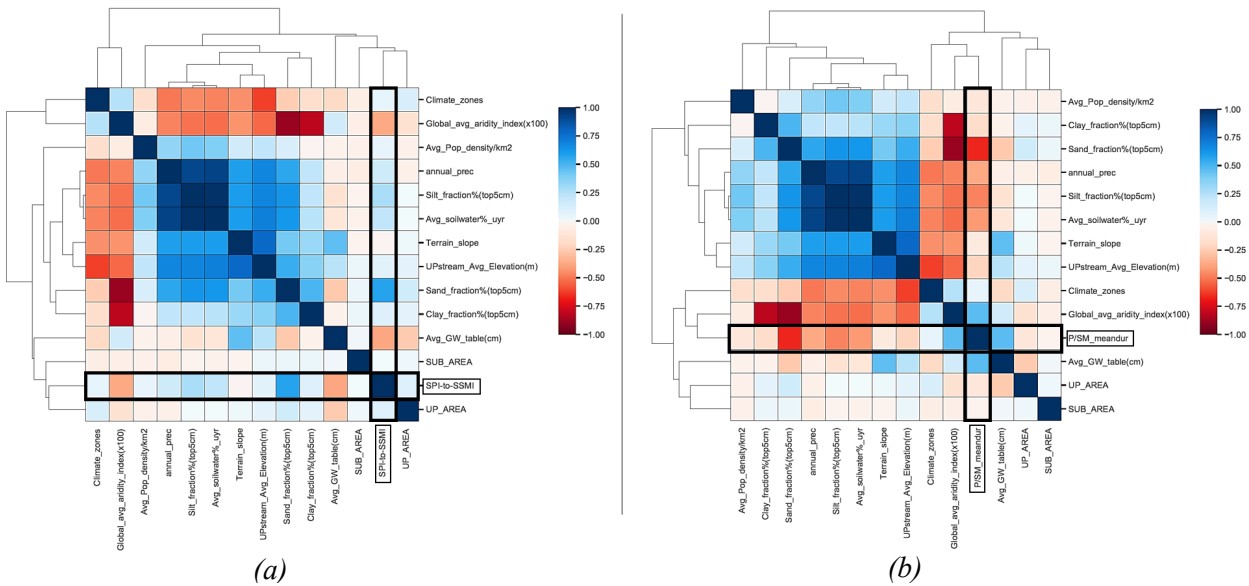

(a)                         (b)

*Figure 4: Heatmap of Pearson correlations between the propagation indices and catchment characteristics: (a) the SPI to SSMI; (b) P/SM mean duration ratio.*
*Note\*. Euclidean distances used for clustering variables with interchangeable correlations. The heatmap based on the Spearman*
*correlation coefficients (see Fig. S11 and S12 Supplementary Material ) showed a similar pattern as Fig. 4. Therefore, we assume that linear models (Pearson correlation method) can be used to represent the monotonic relationships even though the relationships are not perfectly linear.*

    SPI to SSMI propagation has longer accumulation times in catchments with low aridity index and higher sand and low silt content (Table 1) and vice versa. These catchments are located in the (semi-)arid eastern centre of the HOA.

SPI to SSMI propagation also significantly related to percent soil water content and landcover. Similarly, catchments with a low P/SM duration ratio have a low aridity index and higher sand content (Fig. 5a and b). These catchments have low mean annual precipitation and are interspersed with shrubland. They correspond to catchments with slow propagation from meteorological to soil moisture droughts (prolonged soil moisture droughts; Fig. 3d and medium to prolonged meteorological droughts; Fig. 3c) due to the slow response of soil moisture to precipitation. The slow

response is due to the fact that the soil in these areas tends to be very dry, so the soil surface needs to be wetted before infiltration can begin. In addition, SPI to SSMI propagation has a longer accumulation in catchments with closed and open forests and herbaceous wetlands and vegetation (Table 1). In these catchments, the interaction between precipitation and soil moisture is slow, leading to the weaker correlations (Fig. 3a). This phenomenon is consistent with the findings of previous studies (e.g. Sehler et al., 2019), which claimed that landcover, soil moisture and

precipitation are more strongly correlated in (semi-)arid regions with low vegetation, while weaker correlations are found in humid regions with forests and dense vegetation. The propagation of SPI to SSMI has shorter accumulation periods and the P/SM ratio is high in catchments with cropland and bare or sparse vegetation; these catchments are located in the north-western centre (Ethiopian highlands) and north-eastern tip, respectively.

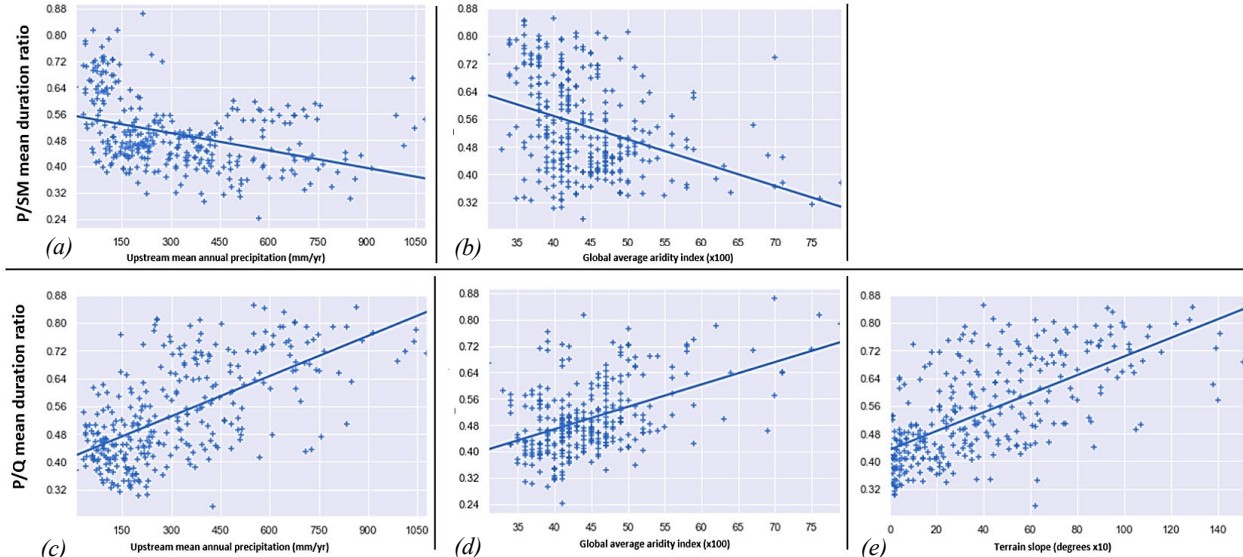


Figure 5: Catchment characteristics against P/SM and P/Q mean duration ratios.

The propagation time from precipitation to soil moisture is also influenced by spatial variability in precipitation within the catchments, but this influence is not pronounced for both SPI-to-SSMI propagation and P/SM ratio. This is reflected in the low positive correlation value between mean annual upstream precipitation and SPI-$n$ for SPI to SSMI
(Fig. 4), the average distribution of accumulation periods per equal interval (quantile) of mean annual upstream precipitation grouping (Table 1), and the less steep slope (highest value 0.56) in P/SM ratio versus mean annual upstream precipitation (Fig. 5a). This weak correlation can be explained by the fact that both the short and long accumulation periods (Fig. 3b) and the low and high P/SM ratios (Fig. 3e) are found in the catchments in the wetter western part of HOA (Fig. 1c). The P/SM ratio decreases with increasing mean annual upstream precipitation, meaning
most catchments with high P/SM ratios in the wetter western part of the HOA respond quickly, while those with low P/SM ratios in the eastern of the HOA respond slowly, they are less affected by precipitation deficits. SPI to SSMI propagation and P/SM ratio are not dependent on upstream area, elevation, and geology (equal distribution of mean values in Table 1) (see Table S2 and S3 Supplementary Material).

Table 1: Mean accumulation period per catchment characteristics: mean annual upstream precipitation, upstream elevation,
upstream area, geological type, and landcover type (≤4 months are considered short accumulation periods and ≥5 long accumulation periods).

| | | SPI to SSMI (months) | SPI to SSI (months) |
|---|---|---|---|
| Mean annual upstream precipitation | *Precipitation (mm)* | | |
| | **10-118** | 6.0 | 3.4 |
| | **118-203** | 5.9 | 2.7 |
| | **203-329** | 4.9 | 2.0 |
| | **329-503** | 4.8 | 1.4 |
| | **503-1079** | 3.1 | 1.0 |
| Upstream Area | *Area (km²)* | | |
| | **152-3935** | 4.3 | 5.6 |
| | **3935-7259** | 4.1 | 5.5 |
| | **7259-15257** | 5.0 | 5.4 |
| | **15257-47890** | 4.3 | 4.4 |
| | **47890-745375** | 4.7 | 4.2 |
| Percent sand fraction | *Sand fraction (%)* | | |
| | **11-26** | 3.1 | 5.5 |
| | **26-29** | 4.4 | 5.6 |

| | | | |
|---|---|---|---|
| | **29-32** | 4.8 | 5.1 |
| | **32-35** | 5.1 | 4.0 |
| | **35-40** | 5.8 | 4.6 |
| | *Elevation (m)* | | |
| Upstream elevation | **3-402** | 3.9 | 6.0 |
| | **402-731** | 4.4 | 6.0 |
| | **732-1043** | 4.7 | 4.9 |
| | **1044-1544** | 5.1 | 4.8 |
| | **1545-2493** | 4.3 | 3.1 |
| | *Geology type* | | |
| Geology | **Crystalline basement** | 5.0 | 3.8 |
| | **Consolidated sedimentary rocks** | 4.4 | 5.0 |
| | **Volcanic rocks** | 4.3 | 3.1 |
| | **Unconsolidated sediments** | 4.4 | 4.4 |
| | **Surface water** | 4.0 | 1.7 |
| | *Landcover types* | | |
| Landcover | **Shrubland** | 4.8 | 4.8 |
| | **Herbaceous vegetation** | 4.6 | 4.2 |
| | **Cropland** | 4.4 | 2.8 |
| | **Built-up** | 4.6 | 3.5 |
| | **Herbaceous wetland** | 4.8 | 2.9 |
| | **Bare/sparse vegetation** | 2.3 | 4.9 |
| | **Open forests** | 5.3 | 3.8 |
| | **Closed forests** | 5.0 | 2.7 |

In summary, the propagation of SPI to SSMI and the P/SM mean duration ratio depend more on the soil properties, landcover and the time of the last rain. All these variables are linked to the storage capacity of the catchment. We see that catchments with a high percentage of sandy soils and shrubland have longer response times and duration, while catchments with a low percentage of sandy soils and cropland have short response times and duration. This link with soil properties is in line with the findings of Van Loon and Laaha (2014), who showed that factors such as storage in soils, aquifers, and lakes influence drought duration with longer durations in larger storage and shorter durations in smaller storage.

## 4.2 Precipitation to streamflow

### 4.2.1 SPI to SSI

The analysis of SPI to SSI (Fig. 6a) shows that the catchments with the low correlation values were mainly found in the north-western centre around the Ethiopian highlands. These areas in the north-western centre also have short accumulation times (Fig. 6c). A few catchments in the north-western tip have longer accumulation periods (5 to 7 months). The majority of catchments in the HOA have long accumulation periods ($\leq$5 months; 212 catchments) and 106 catchments have short accumulation periods ($\geq$ 4 months). Signal strength decreases as it moves down the hydrological cycle, with the highest correlation value being 0.91 for SPI to SSMI and 0.77 for SPI to SSI (Fig. 3a and Fig. 6a, respectively). This is evident in the number of catchments where the strength of correlation value is less than 0.5 (grey sub-catchments in Fig. 6a and c). This results in a smaller number of catchments with correlation values above 0.5, as the drought propagates from meteorological drought to soil moisture to streamflow drought.

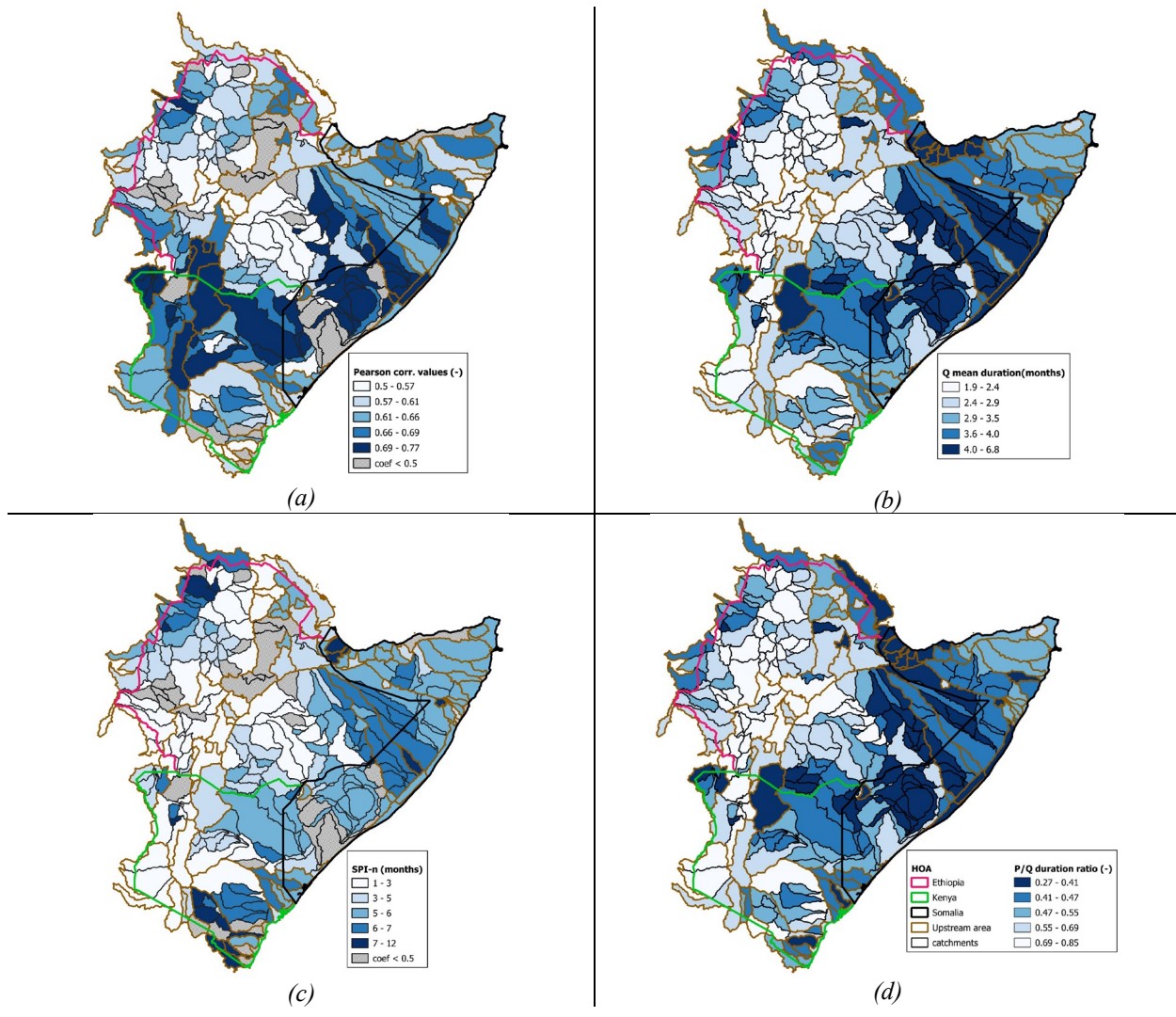

*Figure 6: Propagation from precipitation to streamflow: (a) Highest coefficient values per catchment (>0.5) from SPI to SSI; (b) mean streamflow drought duration; (c) corresponding SPI-n (SPI accumulation period having highest correlation with SSI) per catchment; (d) ratio of mean drought duration from meteorological to streamflow (P/Q ratio).*

### 4.2.2    P/Q mean duration ratio

Analysis of P/Q mean duration ratio (will further be referred to as P/Q ratio) (Fig. 6d) shows that the highest duration ratios are found in catchments to the west and southwest of the HOA, and low ratios are found to the east of the HOA towards the coast. In the catchments with high P/Q ratios, short streamflow droughts and longer meteorological droughts occur, as shown in Fig. 6b and Fig. 3c, respectively. In these locations, the runoff response to rainfall is rapid, in contrast to catchments with low duration ratios that experience longer streamflow droughts, but the streamflow droughts are still longer than meteorological droughts. Streamflow droughts are shorter and there is less pooling of meteorological droughts to soil moisture to streamflow droughts in the west and southwest in contrast to the east towards the coast where they become longer and with increased clustering. The catchments in the east of the HOA are also located within the arid and semi-arid areas. Therefore, whenever it rains, the process of infiltrations has to occur first before any runoff is produced, resulting in longer streamflow to precipitation response. The threshold-based drought duration ratio (P/Q) map (Fig. 6d) shows similar processes to the standardised indices propagation map (Fig. 6c), i.e. short precipitation to streamflow response in the west (represented by short accumulation periods in the standardised indices and a high duration ratio in the threshold indices P/Q ratio).

### 4.2.3 Relation of precipitation to streamflow with climate and catchment characteristics

Propagation of meteorological to streamflow drought is influenced by hydrogeological characteristics of the catchment. In catchments with sedimentary rocks, shrubland, bare or sparse vegetation, low mean annual upstream precipitation, high aridity, low elevation, medium silt content and low sand content, the propagation of SPI to SSI accumulation periods is longer and the P/Q ratio duration of drought is longer. These catchments are located to the east of the HOA towards the coast (Fig. 6c and d) and are associated with small to large upstream catchment areas (Table 1). The influence of upstream areas on propagation was not as pronounced as we expected (Table 1 and Fig. 7). The lack of influence of upstream areas on the propagation of drought from precipitation to streamflow contradicts the findings of previous studies (Haslinger et al., 2014; Van Lanen et al., 2013; Van Loon & Laaha, 2015; Vidal et al., 2010), suggesting that the propagation time from meteorological drought to hydrological drought may be exacerbated by catchment size. In these catchments, the mean duration of streamflow droughts (Fig. 6b) has longer time scales than the mean duration of meteorological droughts (Fig. 3c), reflecting the propagation and suggesting that shorter meteorological drought events are pooled into longer and fewer streamflow drought events due to storage processes in the catchment. Catchment streamflow responded more slowly to precipitation, resulting in longer accumulation periods and low P/Q ratios. This is due to other processes in these areas, such as infiltration and wetting of the soil surface, which have to take place before the runoff occurs. Catchments with short accumulation periods and high P/Q ratios have high mean annual upstream precipitation, low aridity, volcanic soils, cropland, forests and high elevation (Table 1 and Fig. 5). These catchments respond more quickly to precipitation due to the high saturation of the volcanic soils, which are mostly located in the west of HOA. These results are consistent with previous studies (e.g. Li et al. 2019, Laizé & Hannah, 2010) which found that the propagation time from meteorological drought to hydrological drought depends on the flow concentration time, which is strongly influenced by elevation, slope, proportion of cropland and rock permeability.

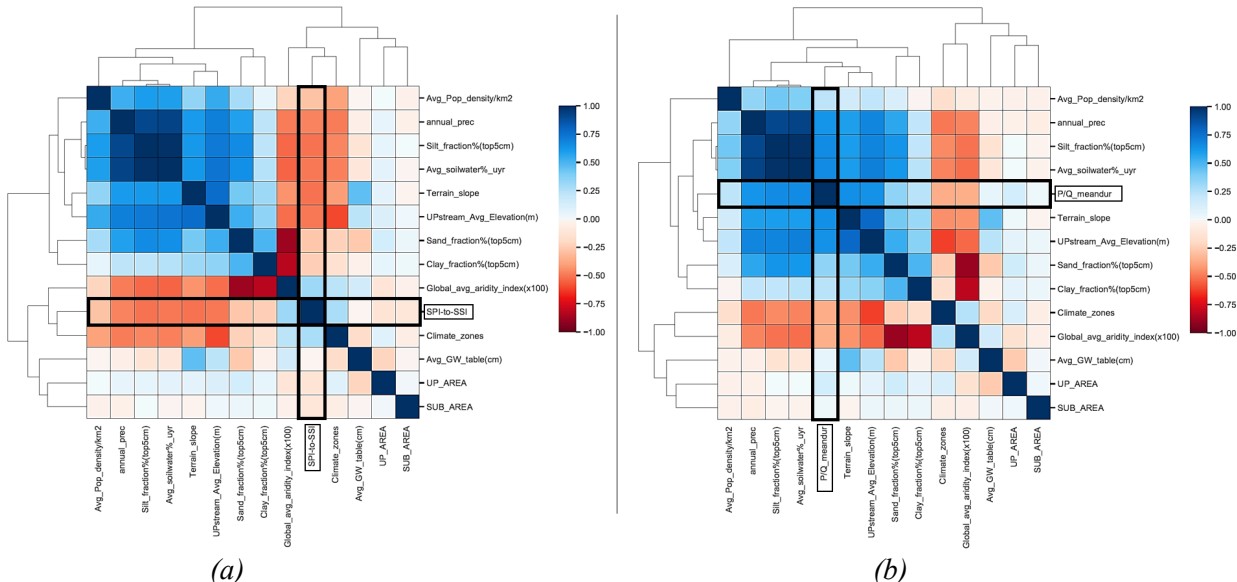

*(a)* *(b)*

*Figure 7: Heatmap of Pearson correlations between the propagation indices and catchment characteristics: (a) the SPI-to-SSI; (b) P/Q mean duration ratio. Euclidean distances used for clustering variables with interchangeable correlations.*

Catchment differences in both SPI to SSI propagation and P/Q mean duration ratios exhibit a spatial pattern that strongly reflects the heterogeneity in geology (Fig. 1d), landcover (Fig. 1a), and precipitation gradient (Fig. 1c) from the wetter west to the drier east of the HOA. The overall precipitation climate of the catchment has a much greater influence on the propagation of meteorological to streamflow drought than on the propagation of meteorological to soil moisture drought (Table 1 and Fig. 5a). The strong link could be due to the prominent precipitation gradient between the humid and the (semi-)arid areas. Rainfall in the semi-arid areas is very erratic and the dry periods last longer, resulting in very low storage (Vicente-Serrano & Lopez-Moreno, 2006) and longer propagation times, which translates into longer droughts. This result is consistent with Van Loon et al. (2014) and Barker et al. (2016) who

found that seasonality of rainfall is an important climatic factor influencing drought propagation of droughts from meteorological to hydrological droughts.

Moreover, the correlation value of mean annual upstream precipitation in SPI to SSI propagation was lower than the value for terrain slope, percent silt content, upstream average elevation and percent average annual soil water content (Fig. 7). This shows that catchment characteristics related to soil properties, geology and landcover have a greater influence on the propagation of drought from meteorological to streamflow drought than the mean annual upstream precipitation. This result is consistent with the findings of Barker et al. (2016), who found that hydrological drought
characteristics of catchments with permeable aquifers have a weak correlation with mean annual precipitation and a strong correlation with catchment storage characteristics such as the Base Flow Index (BFI) or the percentage of highly fractured rock.

# 5    Discussion

## 5.1    Data selection and limitations

For this study, our main objective was to work with observational data, or data that is as close to observations as possible for the entire region. The study utilized MSWEP precipitation data because of its consistency with GLEAM (avoiding accumulation of uncertainty resulting from different sources). The latter estimates soil moisture using satellite imagery and a re-analysis approach. MSWEP precipitation data provides a more accurate and consistent representation of precipitation across Western, Eastern, and Southern Africa. It has shown a strong correlation with
in-situ observations and substantial agreement with Climate Hazards Group InfraRed Precipitation with Station data (CHIRPS), making it a valuable tool for drought monitoring and assessment in the region. CHIRPS data has been popularly applied in the region because it has been found to show a good depiction of rainfall seasonality, and in a study by Musie et al., (2019), they used CHIRPS precipitation to model daily and monthly streamflow and the simulated streamflow data matched streamflow observations). MWEP has better results when compared to ERA-
Interim precipitation data (which was originally applied in the generation of GloFAS river discharge data). These findings are reported in studies by Cattani et al., (2021) and Beck et al., (2017). We chose to not use ERA5 precipitation because the quality is not good and there are no rain-gauge data assimilated into the product.

For discharge, we initially also intended to use observation data or a data product that is as close to observations as possible (like GLEAM for soil moisture). However, the spatial and temporal coverage of observed discharge data in the region is too low for this study. Therefore, we decided to use modelled data for discharge, but there is no dataset
available that uses MSWEP for precipitation input. GloFAS uses ERA5 Land total precipitation data from EMCWF, as input to the hydrological model LISFLOOD. We could have used ERA5 Land precipitation as our precipitation data in this study (for consistency with GloFAS), but we decided against this because ERA5 has been found to highly underestimate/overestimate the precipitation values in the region. Fessehaye et al., (2022) tested the product for Eritrea
region and found it highly underestimated precipitation values. Gleixner et al., (2020), on the other hand, tested the product against CHIRPS dataset and found it overestimated precipitation in East Africa (see Gleixner et al., (2020). GloFAS is calibrated and evaluated against in-situ river discharge, but mainly for perennial rivers at mid-latitudes (Harrigan et al., 2020; Hirpa et al., 2018). When we compared the GloFAS discharge values with GRDC and CETRAD in-situ observations in the study region (with discharge values from 1981 onwards: total of 26 stations), we found that
there often was a strong bias in absolute values (Fig. 8a, b and c), and that the anomalies (value divided by annual mean discharge) are captured well (Fig. 9 a and b stations in Ethiopia). As we work only with relative indices for our drought study (either standardized, or with a relative threshold), the absolute bias is not an issue in our application. Therefore, we decided to use GloFAS data for discharge.

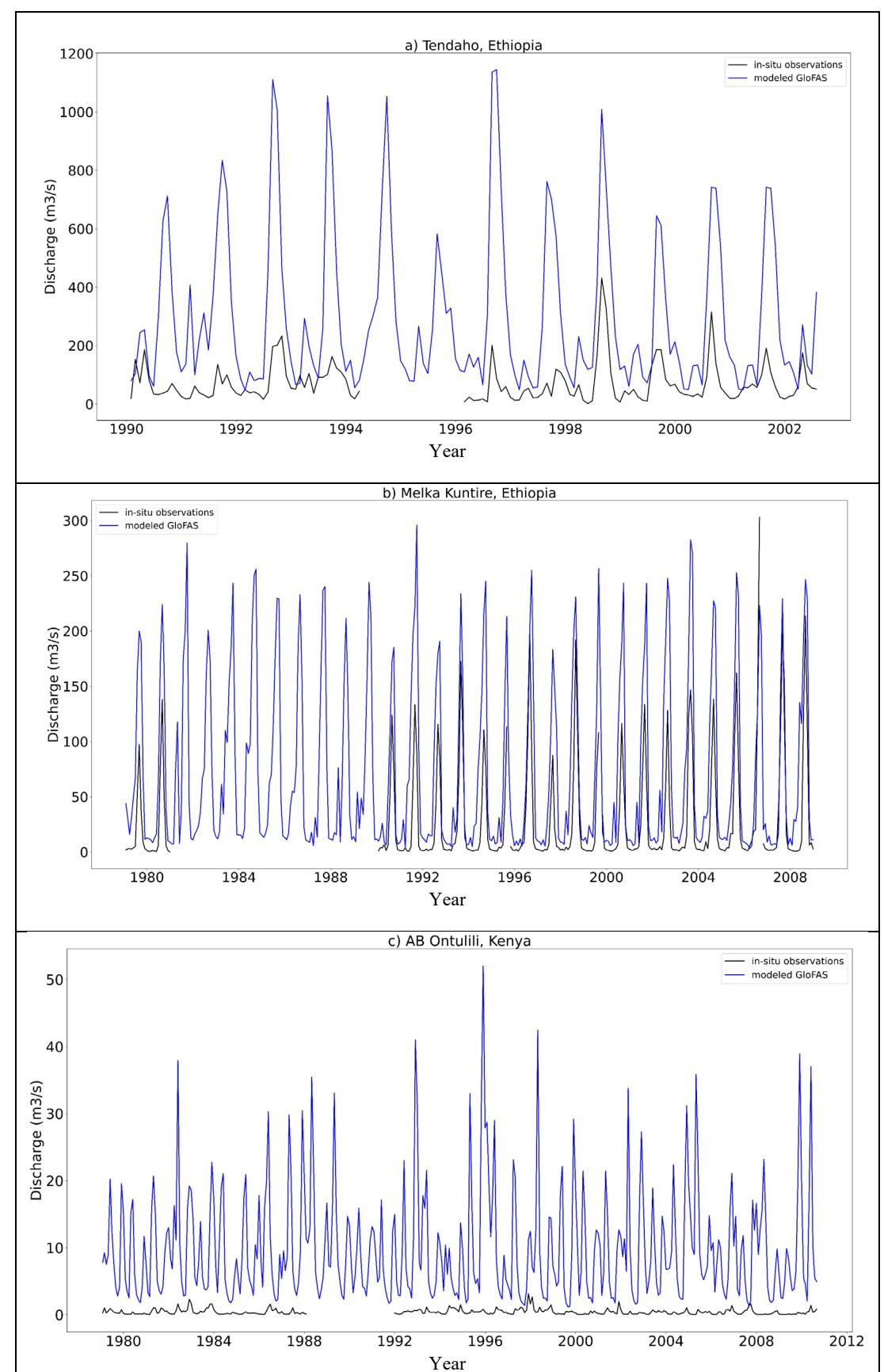

*Figure 8: GloFAS river discharge against in-situ discharge observations in three different gauging stations in HOA; (a) Tendaho gauging station, Ethiopia, (b) Melka Kuntire gauging station, Ethiopia and (c) AB Ontulili gauging station, Kenya.*


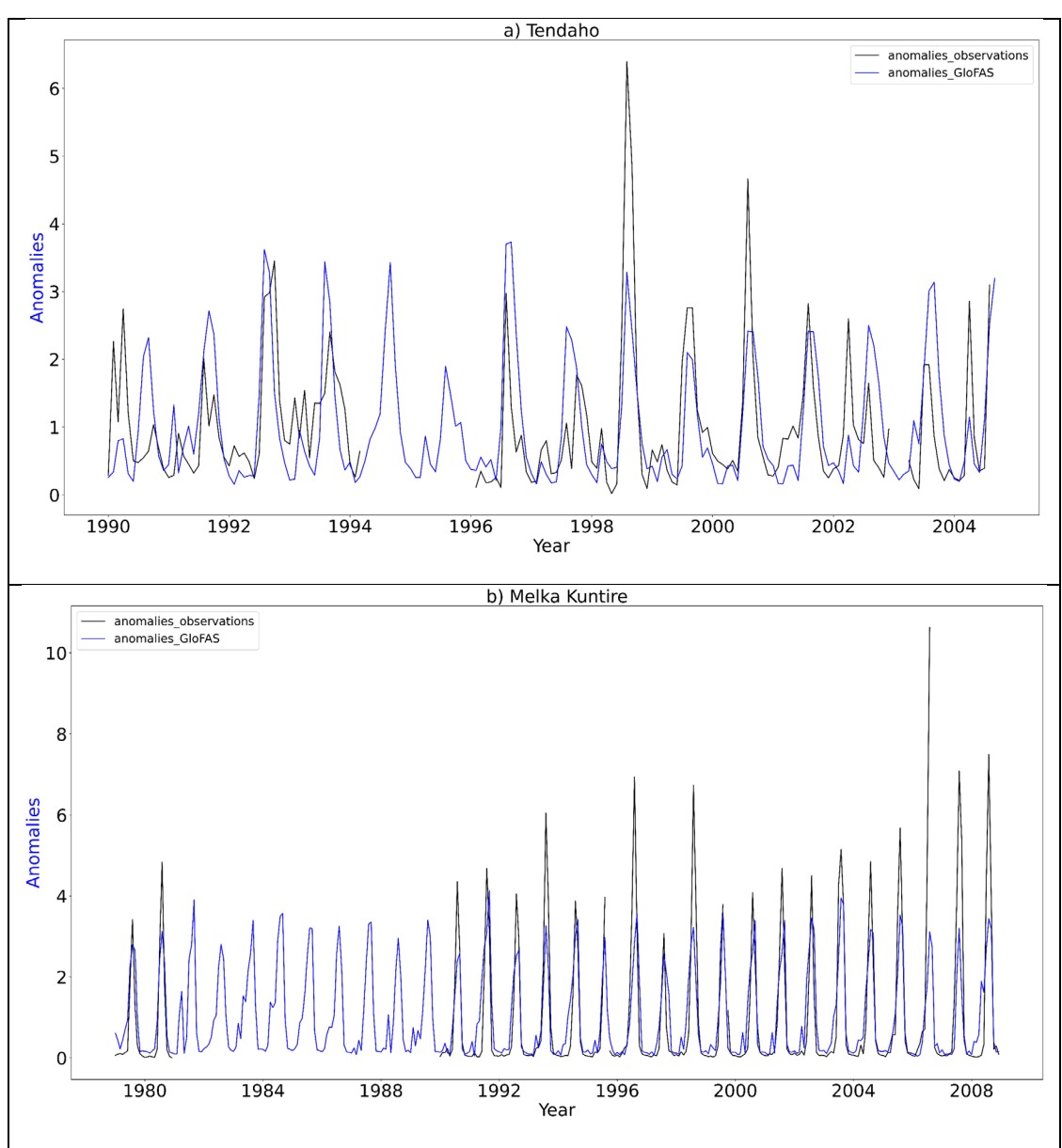

*Figure 9: Discharge anomalies between the observed data and GloFAS streamflow data for two different stations in Ethiopia; (a) Tendaho gauging station and (b) Melka Kuntire gauging station. The deviations are similar.*

The use of appropriate data sources is crucial for accurate modelling and understanding of drought conditions. This study serves as an example of the challenges faced in selecting data sources for regions with limited observed data and highlights the importance of considering multiple factors, including the performance of previous studies and calibration against in-situ observations, when selecting datasets for drought analysis.

## 5.2  Implications for research

When we compare our results with catchment-level studies (see Section 4), we find comparable processes of drought propagation and a similar influence of climate and catchment characteristics on this propagation. As this is a study specific to the HOA region, our results have a number of important implications for drought risk analysis in drylands, but may not be transferable to other regions. The HOA region was chosen because of the high variability in climate and catchment characteristics and the large number of catchments. In the study region, precipitation is related to the elevation (increasing precipitation with increasing elevation and increasing aridity towards the east, where elevation and precipitation also decrease). Comparing catchments in the (semi)-arid region of the study area, it appears that catchment-scale hydrogeological processes, such as geology and landcover, dominate the propagation of drought from meteorological to streamflow, while land surface processes, such as soil properties, influence the propagation from

meteorological to soil moisture drought (Fig. 3). This is consistent with previous studies (e.g. Barker et al., 2016; Van Loon, 2013).

We find that the differences in propagation from meteorological to soil moisture drought are also influenced by spatial variability in precipitation, with the wetter western part of HOA having catchments with both short and long propagation timescales, and the drier eastern having only long propagation timescales. We found that drought are shorter in catchments in the humid west of the HOA (with a higher global average aridity index, fertile volcanic soils and arable land) than in catchments in the (semi-)arid east of the HOA. For example, catchments in the (semi-)arid east with shrubland, bare or sparse vegetation and sedimentary rocks (consolidated and unconsolidated) are affected by longer periods of soil moisture drought. In addition, streamflow drought and the time scale for propagation from meteorological to soil moisture to hydrological drought is also longer (≥5 months). Therefore, for the process of drought propagation from meteorological to streamflow drought, it is important not only to monitor precipitation forecasts but information on hydrogeological characteristics, such as geology and landcover, of the catchment is also essential. Incorporating this knowledge into hydrological drought forecasting could significantly increase the predictive value of forecasting systems by making the forecast less dependent on the predictive skill of actual precipitation. We have also confirmed that drought duration is influenced by both climate and catchment control processes, similar to Van Loon and Laaha, (2014). There is still a need to further investigate the impact on the propagation timescale when we include groundwater.

The study also highlights some of the problems with using SPI, SSMI, SSI and threshold-based duration ratios. Due to the nature of standardised indices, they are unable to identify regions with highly seasonal climates and arid regions (Hayes et al., 1999). Unlike threshold-based indices, it does not capture the water deficits in different catchments and regions that are more prone to drought than others. Moreover, dry regions with the short accumulation periods (1, 2 or 3 months) may have misleadingly high positive and negative values. Although calculating the SPI, SSMI or SSI for any user-defined accumulation periods makes the indicators more flexible, it is still important to choose meaningful accumulation periods to capture drought conditions and also to select appropriate indicators (Vicente-Serrano & Lopez-Moreno, 2005). In addition, the results confirm that accumulation periods should be selected based on the impacts of drought. The threshold-based indices represent the specific duration of drought event, thus better representing the link with catchment characteristics and better capturing seasonal spatial variability in precipitation within the catchments.

## 5.3 Implications for drought monitoring and early warning

Drought mitigation and water resources management require reliable and efficient drought monitoring and early warning systems (M&EW) as they are a critical component of drought preparedness (Barker et al., 2016; Safavi et al., 2018). The efficiency of these systems in analysing extremes is largely determined by the choice of indices, which need to consider and integrate different aspects of information. Drought M&EW systems usually use standardised indices such as the SPI, especially as the SPI is the most widely used index to characterise drought (Vicente-Serrano & Lopez-Moreno, 2005). However, the use of standardised indices and threshold-based indices for soil moisture and hydrological droughts is not widespread or well developed in the HOA. For example, the National Drought Monitoring Agency (NDMA) in Kenya has a good drought M&EW system, but it only considers precipitation, while impacts are more associated with soil moisture and hydrological droughts. The SSI and the use of threshold-based indices are less common in the HOA. This may be due to the lack of streamflow data in this region compared to precipitation data, especially for the short time scales required to produce useful drought M&EW products. However, monitoring soil moisture and hydrological variables and incorporating such indices is beneficial for reliable and effective drought planning and water resource management, and it is particularly useful for communication purposes if precipitation, soil moisture and streamflow are monitored in a comparable manner.

Although the use of streamflow and soil moisture data directly in drought M&EW systems is preferred, these systems cannot be used in this region due to the lack of data. Therefore, the SPI could be a surrogate for soil moisture and streamflow impacts, provided suitable propagation times are known. This also ensures the use of standardised indices in the HOA and discourages the use of threshold-based indices (which require raw data). Given the uncertainty in modelled and reanalysis data, it is better to standardise the datasets, as is the case with standardised indices (Van Loon & Laaha, 2014; Van Lanen et al., 2013; Van Loon, 2013). The correlation results (Fig. 3b and Fig. 6c) showing the spatial variability of SPI-$n$ (the accumulation period strongly correlated with SSMI-1 and SSI-1, respectively) give an indication of accumulation periods that could serve as proxies for soil moisture drought or streamflow drought in the monthly precipitation data. This allows the use of precipitation data that is more readily available to identify future

potential soil moisture and streamflow droughts. In addition, the short soil moisture and streamflow droughts, which are more influential for drought planning and water resource management, are better captured by the short accumulation periods (Fig. 3b and Fig. 6c), which are less affected by decreasing long-term trends in precipitation and streamflow in the eastern HOA and increasing trends in the western HOA (Gebrechorkos et al., 2020). Water managers can use this information on soil moisture and streamflow trends to identify when to begin controlling water users and anticipate drought impacts. The results obtained can be used to forecast water resource.

## 5.4 Recommendations and further research

Groundwater plays an important role in mitigating the impacts of drought and as a source of water supply in arid and semi-arid areas, especially in the eastern part of the HOA (Adloff et al., 2022). Therefore, to fully understand the process of drought propagation, it is necessary to include the groundwater component in the analysis. Furthermore, while catchment storage plays a key role in determining the drought duration and propagation, it is also important to consider seasonality and autocorrelation of soil moisture, as well as streamflow caused by infiltration and evaporation. Therefore, analysing the propagation of the drought signal through the hydrological cycle and including the groundwater component would provide a more comprehensive picture and assessment of the influence of climate and catchment characteristics on the duration, severity and propagation. In addition, the impact of seasonal variability (based on long and short rains) on drought propagation should be further investigated. Seasonal variability is particularly important for the propagation from meteorological to soil moisture drought, especially in the western part of the HOA where the soil moisture response to precipitation depends on when it last rained. Similarly, the timing of hydrological droughts leading to impacts should be investigated.

Finally, the availability of hydrological records for observation-based studies of drought is a limitation. This is particularly true for the HOA. The period of analysis (1980-2020) does not capture the full range of hydrological variability. We anticipate that longer records could affect the accumulation periods presented here, although the same regional picture and propagation characteristics would likely emerge. In addition, the use of modelled and reanalysis data has introduced some uncertainty into the analysis. For example, the GloFAS streamflow dataset was developed for a global application and represents streamflow in perennial systems typical of humid regions. Accordingly, it does not represent ephemeral flow processes typical in dry regions. The dataset tends to over-/underestimate the streamflow in arid and semi-arid areas. Therefore, a modelling framework suitable for (semi-)arid areas, where hydrological processes differ from those in humid regions is crucial. For example, a model such as the DRYP hydrological model (Quichimbo et al., 2021) has been developed specifically for hydrological processes such as ephemeral flow, surface and groundwater interactions, and high resolution precipitation in (semi-)arid regions, and therefore has good potential for further investigation and application. This model has been used to investigate the role of gridded precipitation resolution in socially relevant water stores (streamflow, soil moisture and groundwater recharge) and has been used to make water balance predictions based on seasonal climate projections in the HOA. A regional version of this model would provide a better alternative for follow-up studies given the GloFAS dataset limitations.

## 6 Conclusion

Drought propagation from meteorological to soil moisture to hydrological drought in 318 catchments in the HOA was analysed using standardised indices (over a range of accumulation periods) and threshold-based indices (drought-duration ratios). In addition, the influence of possible governing factors, such as climate and catchment characteristics, was also investigated. The research shows that:

- Precipitation to soil moisture propagation time is longer (5 to 7 months) in catchments with shrubland, closed and open forests, herbaceous wetland and vegetation, and high sand and low silt fraction, while being shorter (2-4 months) in catchments with cropland and high mean annual upstream precipitation.
- Precipitation to streamflow propagation time is longer in catchments with sedimentary rock structure, low mean annual precipitation, and shrubland, while being shorter in catchments with volcanic soils, high annual mean precipitation, cropland and forests.
- In precipitation to streamflow propagation the catchment properties related to soil properties, geology, elevation and landcover are more influential than mean annual upstream precipitation. However, the mean annual upstream precipitation is not so important for streamflow drought duration, and propagation from precipitation-to-streamflow; but nevertheless mean annual upstream precipitation is even less important in propagation from precipitation to soil moisture .

In summary, precipitation to soil moisture propagation is more dependent on the soil properties as opposed to the hydrogeological characteristics (i.e., elevation) while the precipitation to streamflow propagation experience the combined effect of climate and catchment control properties (i.e., elevation, geology). The results in this study provide an indication of precipitation accumulation periods that could serve as a proxy for soil moisture and streamflow droughts in the HOA. The precipitation accumulation periods of roughly 1 to 4 months in wet western areas of HOA, and of roughly 5 to 7 months in the more dryland regions are the most suitable for drought analysis. These results can be used as a foundation for future developments in drought monitoring and early warning systems in the HOA, laying foundations for better drought preparedness and increased resilience to drought and its impacts in water resources.

*Author contributions*. RAO performed the analysis and wrote the paper. HdM and AFvL participated in technical discussions and co-wrote the paper.
*Competing interests*. The authors have no competing interests to declare.

*Acknowledgements*. This study is an outcome of the Down2Earth Project (D2E). An EU Horizon 2020 Project funded under grant agreement number 869550. We thank Marthe Wens (Institute of Environmental Studies, Vrije Universiteit Amsterdam) for help with python scripts. The authors would like to thank Bristol Principal Investigator on D2E Katerina Michaelides (University of Bristol, UK) for her constructive comments that helped improve the paper.

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
