# Peer review of "Propagation from meteorological to hydrological drought in the Horn of Africa using both standardised and threshold-based indices"

_EGUsphere, 2022_

## Referee Comment (RC1)

**Propagation from meteorological to hydrological drought in the Horn of Africa using both standardized and threshold-based indices**

The study analyzes the propagation of meteorological drought to soil moisture and hydrological drought and their influencing factors in the Horn of Africa (HOA) covering Kenya, Somalia and Ethiopia and consisting of 338 catchments. In this study, the authors present a variety of gridded datasets for analyzing droughts at a monthly level during the period 1980–2020, including MSWEP precipitation, GLEAM soil moisture and GloFAS streamflow. As a first step, they calculate the standardized and threshold-based indices and examine propagation from one to another.

The main contribution of the paper is its effort at proposing a methodological framework, while relying on well-known approaches and methods. I suggest some improvements detailed below. I hope my comments can contribute to enhancing the quality of the paper.

**Major comments**

- The first thing I would suggest to the authors is that they carefully review the text to avoid several grammar errors and typographical errors prevalent in the manuscript (I list some of these errors at the end of the review as examples).
- Studies have developed and used a wide range of meteorological drought indices. Could authors briefly explain why they selected/prefer SPI over other indices?
- Authors did not explain/mentioned how they identify drought events based on these standardized indices (onset and termination of drought events). Since these standardized indices encompass both droughts and non-drought periods.
- Is there any particular reason for having different interval classes in Figure 1b? If not, then I would suggest that the class interval and color scale be changed with distinct colors for the classes.
- Figure 2: The authors mentioned two drought characteristics i.e., duration and severity/deficit volume. However, the entire paper focuses exclusively on the duration of the drought. Therefore, it is recommended that the severity/deficit volume analysis be added or removed from Figure 2.

**Minor Comments**
- It is preferable to have different color boundaries for the countries so that the location of each country can be identified immediately.
- Figure 2: The resolution of the figure is quite low
- I believe this paragraph should be placed in the introduction rather than on lines 124-132
- Lines 209-211: It is appropriate to provide these results as supplementary information.

**Typos and English grammar (examples)**

- Line 15-16: "and by calculating the ratio between the threshold respectively streamflow drought duration". The use of respectively is not clear, please rephrase
- Line 425: We find differences in propagation from precipitation to soil moisture to also be influenced by ….
- Line 509: repetition of the sentence (As such, the dataset tends to overestimate streamflow in arid and semi-arid areas).

---

## Referee Comment (RC2)

I have reviewed the manuscript egusphere-2022-1307 "Propagation from meteorological to hydrological drought in the Horn of Africa using both standardized and threshold-based indices" which illustrates an assessment of drought propagation, from meteorological to soil moisture and hydrological drought, in the Horn of Africa by using well known approaches.

First of all I would congratulate with the authors on the very important research effort, given the large number of catchments studied and the peculiarity of the region, including the difficulties in gathering the data. As highlighted by the same authors in the introduction, the novelty aspect introduced is mainly represented by the case study which suffers, as in many other areas of the world, from a lack of information and data available for evaluation. However, there are several major (and minor) comments that should be addressed before the publication stage.

Major points to be addressed:
1) First of all, the question of the uncertainty relating to the data used should be addressed. Since the research work deals with modeled, re-analyzed and gridded data, coming from different temporal and spatial aggregation scales and from different models, each data has its own uncertainty and in their combined use it is not possible to verify this question, not even the different uncertainties can be compared each other. GLEAM for example uses MSWEP as input, but what can we say about GloFAS? What rainfall input does it use? If different from MSWEP, how can we compare the modeled flow data with the rainfall data of MSWEP? Another source of uncertainty is also probably due to the fact that the different standardized indices (SPI, SSMI, SSI) are calculated for each basin with reference to different probability laws. How this would have impacted the analysis?
2) I understand that the research idea is the propagation of drought, but perhaps a risk analysis (see Figure 2), which also showed the quantitative evaluation and the relative characteristics of the indices, also from a spatial point of view, would have helped to sort out the problem. For example, on line 443 it is said, if I understand correctly, that the drought indices used "fail to capture the water deficit amount…" how can we know this if we do not have an illustration of the quantitative assessment of the indices?
3) In general, the quality of preparation should be improved. Some sections of the manuscript were particularly difficult to read and sometimes repetitive. For example, section 3.3 says little more than 3.3 so maybe they could be merged. In general, the paragraph on methodology should be improved. In the same way, the presentation of figures (figure 5) and tables (table 1) in points of the body of the text where the analyses have not yet been inserted are misleading.
4) In general, even the figures should be improved in their quality and also in the formatting of the characters, which even when printed are too small for easy reading. I wonder why in figure 3 the same colours are always used except for panel (a) and the same for figure 6. Why don't change them or just have always the same colour? In figure 3 and 6 there are no units of measurement. Furthermore, in table 1 it is not clear to me (not even from the caption) what is the difference between the two SPI-to-SSMI (months) columns. The two columns have the same header but different values.

Other minor points to be addressed:

1) Line 94: what is "level 6 boundaries of HydroBASINs?
2) Line 110: You probably selected the study region not because it is an "interesting" region but because, given the large number of catchments and relevant catchments attributes, it would have been suitable for the purpose of the study.
3) Line 116: "followed by the calculation of the indices". Which indices?
4) Line 146: "three hourly temporal", please clarify
5) Line 169: what "HAD" is?

6) Line 209: it is questionable the fact you selected 70$^{th}$ percentile because otherwise you have a too small number of events. The selection of the percentile deals with the severity of the drought events. Probably another message would have come from the analysis.

7) Line 215-218: it is not clear to me the explanation provided when P/Q is close to zero. This should happen when the duration of the meteorological drought is significantly smaller than the one for the hydrological drought, and this would happen when you have a small number of meteorological drought events or short meteorological drought events. Also I believe authors should better explain why they decided to use P/Q and P/SM ratio as indicators of drought propagation, what does this ratio means and why they did not use more convention drought propagation index (line 233 it is said that you did not consider the lag because in some previous study it is said that the largest correlation occurs at lag 0)

8) Line 242-243: reformulate

9) In my opinion figure 3 panel (b) and do not properly give the same message, much better in the case of streamflow (figure7).

10) Line 319: in my opinion it is improper to discuss about "precipitation variability" as you only considered the mean annual precipitation for each catchment and this is not a variability index for precipitation (also on line 425)

11) Paragraph 5.1 and 5.2 present different obvious sentences, not really of interest for the research (lines 451-456, …)

---

## Author Comment (AC1)

**#Reviewer 1-RC1**

We want to thank referee 1 for the critical review of our manuscript and for the positive words about our paper and its contribution. We have reflected on the comments and below are our point-to-point response (in bold, with the original reviewer comment in normal format) to the questions raised.

*Major comments*

- The first thing I would suggest to the authors is that they carefully review the text to avoid several grammar errors and typographical errors prevalent in the manuscript (I list some of these errors at the end of the review as examples).

**We agree. We will carefully review the manuscript text for the suggested grammatical errors and any other errors we may have missed.**

- Studies have developed and used a wide range of meteorological drought indices. Could authors briefly explain why they selected/prefer SPI over other indices?

**We preferred to use SPI over other indices because we learned from the National Drought Management Authority of Kenya, that they used SPI specifically for the monitoring of drought in the country. This is also like the other organizations delivering climate services to Horn of Africa like IGAD Climate Prediction and Application Centre (ICPAC) in their East Africa Drought Watch. Several studies in the Horn of Africa have also applied SPI. In our study, we do not only use standardized indices like SPI, we also compare SPI to threshold-based indices and compare how standardized and threshold-based indices characterize drought propagation in the region. We will clarify this in the revised version**

- Authors did not explain/mentioned how they identify drought events based on these standardized indices (onset and termination of drought events). Since these standardized indices encompass both droughts and non-drought periods.

**In the analysis, we tested two separate methods. The first was the standardized indices method which included standardized wet and dry periods to characterize *drought months*. As such we did not identify drought events using standardized indices, but rather the relative dryness over different accumulation periods without defining whether it's a drought event or not. The second method was the threshold-based indices which we applied a threshold (70th percentile) to characterize *drought events* (Heudorfer and Stahl, 2017). So here we do identify events and define it in terms of duration. We applied a variable threshold (without pooling) which changed each month over the years. Each of these methods highlighted different aspects of the drought. The threshold-based duration preserved the original hydrological values and estimated how long the water shortage lasted while the standardized indices failed to preserve the hydrological values, but no (arguably arbitrary) threshold had to be used. However, both methods do say something about the number of dry months, but in different ways: either as duration of an event, or the accumulation period. We will explicitly mention this in our explanation in the Methods section.**

- Is there any reason for having different interval classes in Figure 1b? If not, then I would suggest that the class interval and color scale be changed with distinct colors for the classes.

**There was no reason for this. We did this because the list of elevation values were too many and we needed to try to display all the classes, hence the use of intervals. We agree to change the color scheme as suggested.**

- Figure 2: The authors mentioned two drought characteristics i.e., duration and severity/deficit volume. However, the entire paper focuses exclusively on the duration of the drought. Therefore, it is recommended that the severity/deficit volume analysis be added or removed from Figure 2.

**We agree these parts should be removed from Figure 2.**

*Minor Comments*

**We will address all the below minor comments in the revised version of the manuscript**

- It is preferable to have different color boundaries for the countries so that the location of each country can be identified immediately.
- Figure 2: The resolution of the figure is quite low
- I believe this paragraph should be placed in the introduction rather than on lines 124-132
- Lines 209-211: It is appropriate to provide these results as supplementary information.
- Typos and English grammar (examples)
- Line 15-16: "and by calculating the ratio between the threshold respectively streamflow drought duration". The use of respectively is not clear, please rephrase
- Line 425: We find differences in propagation from precipitation to soil moisture to also be influenced by ….
- Line 509: repetition of the sentence (As such, the dataset tends to overestimate streamflow in arid and semi-arid areas).

**REFERENCE**

Heudorfer, B. and Stahl, K.: Comparison of different threshold level methods for drought propagation analysis in Germany, Hydrol. Res., 1311–1326, https://doi.org/10.2166/nh.2016.258, 2017.

---

## Author Comment (AC2)

We want to thank referee 2 for the critical review of our manuscript and for the positive words about our paper and its contribution. We have reflected on the comments and below are our point-to-point response to the questions raised.

*Major comments*

- First of all, the question of the uncertainty relating to the data used should be addressed. Since the research work deals with modeled, re-analyzed and gridded data, coming from different temporal and spatial aggregation scales and from different models, each data has its own uncertainty and in their combined use it is not possible to verify this question, not even the different uncertainties can be compared each other. GLEAM for example uses MSWEP as input, but what can we say about GloFAS? What rainfall input does it use? If different from MSWEP, how can we compare the modeled flow data with the rainfall data of MSWEP?

The source and quality of the data we use has indeed been a topic we discussed a lot and deserves more explicit inclusion in the manuscript. Our main objective was to work with observational data, or data that is as close to observations as possible for the entire region. For precipitation there were various good products available. We chose for MSWEP because that is consistent with GLEAM (avoiding accumulation of uncertainty resulting from different sources), which in turn estimates soil moisture using satellite imagery and a re-analysis approach. The MSWEP precipitation data was also selected due to its demonstrated good performance across Western, Eastern, and Southern Africa. It has shown a strong correlation with in-situ observations and substantial agreement with CHIRPS precipitation data (has been popularly applied in the region because it has been found to show a good depiction of rainfall seasonality, and in a study by Musie et al., (2019), it was also found to capture daily and monthly streamflow simulation). MWEP has better results when compared to ERA-Interim precipitation data (which was originally applied in the generation of GloFAS river discharge data). These findings are reported in studies by Cattani et al., (2021) and Beck et al., (2017). We chose to not use ERA5 precipitation because the quality is not good and there are no rain-gauge data assimilated into the product.

For discharge, we initially also wanted to use observation data or a data product that is as close to observations as possible (like GLEAM for soil moisture). However, the spatial and temporal coverage of observed discharge data in the region is too low for our analysis. Therefore we decided to use modelled data for discharge, but there is no dataset available that uses MSWEP for precipitation input. GloFAS uses ERA5 Land total precipitation data from EMCWF, distributed by the hydrological LISFLOOD model. We did not want to use ERA5 Land precipitation because it has been found to highly underestimate/overestimate the precipitation values in the region. Fessehaye et al., (2022) tested the product for Eritrea region and found it highly underestimated precipitation values. Gleixner et al., (2020), on the other hand, tested the product against CHIRPS dataset and found it overestimated precipitation in East Africa (see a copy of figure 3). GloFAS is calibrated and evaluated against in-situ river discharge, but mainly for perennial rivers at mid-latitudes (Harrigan et al., 2020; Hirpa et al., 2018). When we compared GloFAS dataset with GRDC in-situ observations in the study region, we found that there often was a strong bias in absolute values (see figure 1), but that the anomalies (value divided by annual mean discharge) are much better captured (see below figure 2 for two different stations in Ethiopia). As we mainly work with relative indices for our drought study (either standardized, or with a relative threshold), the absolute bias is less of an issue in our application so we decided to use GloFAS. We will suggest

**that we include these figures in supplementary material and make our reasons for the use of GloFAS and uncertainty generated more explicit in manuscript.**

[Figure]

*Fig. 1: Plots of GloFAS river discharge against observations*

[Figure]

[Figure]

*Fig. 2: Plots showing the discharge anomalies between the observed data and GLoFAS discharge. The deviations are similar as stated above*

[Figure]

*Fig. 3: Plots of ERA5 Land precipitation against CHIRPS dataset (Glexner et. al., 2020)*

- Another source of uncertainty is also probably due to the fact that the different standardized indices (SPI, SSMI, SSI) are calculated for each basin with reference to different probability laws. How this would have impacted the analysis?

We indeed applied different distributions and picked the one with the best fit to calculate each of the indices. By doing this, we are not consistent between catchments, which cause uncertainties when comparing values between different catchments. As we are mainly looking to compare within catchments, we prefer the better fits to the spatial consistency. However, we only compare standardized values within catchments.

By the nature of the different indices, different distributions are best suited to fit the different data types. This is recognized in literature and as such we used the distributions suggested by Stagge et al., (2015) for calculation of SPI, distributions suggested by Ryu and Famiglietti, (2005) for calculation of SSMI and distributions suggested by Vicente-Serrano et al., (2012) for calculation of SSI. [we will explain in manuscript]

We agree that fitting a different distribution for each catchment causes some uncertainty, especially in the tails (i.e. drought extremes). However, this is only relevant if one is directly comparing drought onset or severity of the same event between catchments. In our case we do not analyze the droughts themselves, but instead we look at the propagation in the hydrological system. Therefore, we correlated the time series of the meteorological indices with several accumulation periods to the hydrological indices to fit the best correspondence. In our opinion, in this case the benefits of having the best fit to the data for each catchment outweigh the differences in extreme droughts between catchments.

Another way to verify our results is the comparison with threshold-based indices. Our propagation analysis done with the threshold-based duration ratios is very similar to that based on accumulation periods of standardized indices. This indicates that the distribution fitting is expected to have a minor role.

To give the reviewer more details about the distribution fitting, we here include some extra analysis. For example looking at the fitting of precipitation data for calculation of SPI-1 for the different catchments, we can see in the figures below that the depiction of the drought months is not different. This was similar for the other indices.

[Figure]

*Fig. 4: Histogram showing the number of times each distribution was used in the calculation of SPI-1*

[Figure]

*Fig. 5: shows the fitting of the distribution for the calculation of SPI-1-> selected best distribution is framed in black*

**The fittings for the distributions in the catchments in general were not very good as in figure 5, hence we decided to not force fit data for each catchment into the same distribution to avoid introducing artifacts. This is because if we do force fit a consistent distribution throughout the catchments, especially in dry catchments with lots of zero values, we anticipate worser fitting distribution and even worser results. Additionally, we are rather more interested in the catchment propagation than the spatial pattern existing among the catchments, because we would like to better identify the droughts accurately within each catchment.**

**See below plots for four different catchments.**

[Figure]

[Figure]

[Figure]

[Figure]

*Fig. 6: shows the SPI-6 after fitting different distribution through the precipitation data for different catchments->selected best distributions are framed in black*

**There was a similar representation when looking at the SSMI-3 distribution fitting. There is very minor difference in the distributions (see below plot). Therefore, we concluded that the use of the different distributions will not affect the results in the long run. We suggest that we add a few lines about this in the discussion of the manuscript.**

[Figure]

*Fig. 7: shows the SSMI-6 after fitting different distribution through the soil moisture data ->selected best distribution is framed in black*

- ▪ I understand that the research idea is the propagation of drought, but perhaps a risk analysis (see Figure 2), which also showed the quantitative evaluation and the relative characteristics of the indices, also from a spatial point of view, would have helped to sort out the problem. For example, on line 443 it is said, if I understand correctly, that the drought indices used "fail to capture the water deficit amount…" how can we know this if we do not have an illustration of the quantitative assessment of the indices?

**Line 443 was based on a literature review on the advantages and disadvantages of using standardized indices against the threshold-based indices. Which index would be better able to capture absolute water deficits was not tested in this manuscript. Looking into drought characteristics themselves and their spatial representation was beyond the scope of our analysis. It would have made the manuscript substantially longer and would have taken away the focus from the propagation results, which in our view are more interesting to study. Additionally, as we discussed in our response to the previous comment, the spatial comparison of the drought**

characteristics themselves has large uncertainties due to the fitting of different distributions in each catchment. We do agree to include the plots in the supplementary material for different catchments located in different places within the region of study for the readers, but would be hesitant to draw any conclusions from these.

- In general, the quality of preparation should be improved. Some sections of the manuscript were particularly difficult to read and sometimes repetitive. For example, section 3.3 says little more than 3.3 so maybe they could be merged. In general, the paragraph on methodology should be improved. In the same way, the presentation of figures (figure 5) and tables (table 1) in points of the body of the text where the analyses have not yet been inserted are misleading.

**We will carefully read through the manuscript and remove repetitions while also merging the suggested sections.**

- In general, even the figures should be improved in their quality and also in the formatting of the characters, which even when printed are too small for easy reading. I wonder why in figure 3 the same colours are always used except for panel (a) and the same for figure 6. Why don't change them or just have always the same colour? In figure 3 and 6 there are no units of measurement. Furthermore, in table 1 it is not clear to me (not even from the caption) what is the difference between the two SPI-to-SSMI (months) columns. The two columns have the same header but different values.

**In Figure 3a represents the correlation values obtained during the propagation analysis while Figures 3b-e represent propagation indices maps, hence, the different colours. Same explanation applies to Figure 6. For this reason, we suggest to keep the same colour scheme in both figures for differentiation purposes.**

**We agree to include the units of measurements in both figures. We did try the suggested format for table 1 but realized this made it harder to read.**

**In Table 1, the different values are for each of the climate and catchment characteristics (the table has been split into 4 columns to reduce the length) . The values represent the mean aggregation of the selected indices of each catchment per each class of the climate and catchment characteristics. This was done to show the average selection in months of each of the indices per class. We agree to include a similar explanation in the manuscript for better understanding.**

*Minor comments*

- Line 94: what is "level 6 boundaries of HydroBASINS?

**HydroBASINS is a system that divides a large water basin into smaller sub-basins at points where two river branches meet and each has a minimum upstream area of 100 km². The sub-basins are also grouped and coded to allow for the creation of smaller sub-basins at different scales, or to move from upstream to downstream within the sub-basin network. To make this possible, HydroBASINS uses the "Pfafstetter" coding system, offering 12 nested sub-basin levels globally. We will expand the explanation of the selected level in the manuscript.**

- Line 110: You probably selected the study region not because it is an "interesting" region but because, given the large number of catchments and relevant catchments attributes, it would have been suitable for the purpose of the study.

**We agree and we will give a better reason behind the selection of the study region.**

- Line 116: "followed by the calculation of the indices". Which indices?

**SPI, SSMI,SSI and drought duration indices. We will add.**

- Line 146: "three hourly temporal", please clarify

**The MSWEP precipitation data has a resolution of 0.1 degrees grid with 3 hour intervals. Though for the analysis we used the daily precipitation data.**

- Line 169: what "HAD" is?

**This should have been Horn of Africa (HOA). We will change.**

- Line 209: it is questionable the fact you selected 70th percentile because otherwise you have a too small number of events. The selection of the percentile deals with the severity of the drought events. Probably another message would have come from the analysis

**We agree to rephrase.**

- Line 215-218: it is not clear to me the explanation provided when P/Q is close to zero. This should happen when the duration of the meteorological drought is significantly smaller than the one for the hydrological drought, and this would happen when you have a small number of meteorological drought events or short meteorological drought events. Also I believe authors should better explain why they decided to use P/Q and P/SM ratio as indicators of drought propagation, what does this ratio means and why they did not use more convention drought propagation index (line 233 it is said that you did not consider the lag because in some previous study it is said that the largest correlation occurs at lag 0)

**We agree to give a more detailed description and explanation in the manuscript detailing the reasons for the use of the duration ratios and their representation.**

- Line 242-243: reformulate

   **We agree to rephrase.**

- In my opinion figure 3 panel (b) and do not properly give the same message, much better in the case of streamflow (figure7).

**We do not understand what the reviewer meant.**

- Line 319: in my opinion it is improper to discuss about "precipitation variability" as you only considered the mean annual precipitation for each catchment and this is not a variability index for precipitation (also on line 425)

**We use mean annual precipitation as a descriptor of the precipitation climate as has been used in previous studies (Barker et. al., 2016) and to refer to the spatial differences in precipitation. We agree to avoid using this term.**

- Paragraph 5.1 and 5.2 present different obvious sentences, not really of interest for the research (lines 451-456, …)

**We agree to remove the paragraph**

**REFERENCES**

Beck, H. E., Vergopolan, N., Pan, M., Levizzani, V., van Dijk, A. I. J. M., Weedon, G., Brocca, L., Pappenberger, F., Huffman, G. J., and Wood, E. F.: Global-scale evaluation of 23 precipitation datasets using gaugeobservations and hydrological modeling, Global hydrology/Instruments and observation techniques, https://doi.org/10.5194/hess-2017-508, 2017.

Cattani, E., Ferguglia, O., Merino, A., and Levizzani, V.: Precipitation Products' Inter–Comparison over East and Southern Africa 1983–2017, Remote Sens., 13, 4419, https://doi.org/10.3390/rs13214419, 2021.

Fessehaye, M., Franke, J., and Brönnimann, S.: Evaluation of satellite-based (CHIRPS and GPM) and reanalysis (ERA5-Land) precipitation estimates over Eritrea, Meteorol. Z., 31, 401–413, https://doi.org/10.1127/metz/2022/1111, 2022.

Gleixner, S., Demissie, T., and Diro, G. T.: Did ERA5 Improve Temperature and Precipitation Reanalysis over East Africa?, Atmosphere, 11, 996, https://doi.org/10.3390/atmos11090996, 2020.

Harrigan, S., Zsoter, E., Alfieri, L., Prudhomme, C., Salamon, P., Barnard, C., Cloke, H., and Pappenberger, F.: GloFAS-ERA5 operational global river discharge reanalysis 1979 present, GloFAS-ERA5 Oper. Glob. River Disch. Reanalysis 1979- Present, 1–23, https://doi.org/10.5194/essd-2019-232, 2020.

Hirpa, F. A., Salamon, P., Beck, H. E., Lorini, V., Alfieri, L., Zsoter, E., and Dadson, S. J.: Calibration of the Global Flood Awareness System (GloFAS) using daily streamflow data, J. Hydrol., 566, 595–606, https://doi.org/10.1016/j.jhydrol.2018.09.052, 2018.

Musie, M., Sen, S., and Srivastava, P.: Comparison and evaluation of gridded precipitation datasets for streamflow simulation in data scarce watersheds of Ethiopia, J. Hydrol., 579, 124168, https://doi.org/10.1016/j.jhydrol.2019.124168, 2019.

Ryu, D. and Famiglietti, J. S.: Characterization of footprint-scale surface soil moisture variability using Gaussian and beta distribution functions during the Southern Great Plains 1997 (SGP97) hydrology experiment, Water Resour. Res., 41, https://doi.org/10.1029/2004WR003835, 2005.

Stagge, J. H., Kohn, I., Tallaksen, L. M., and Stahl, K.: Modeling drought impact occurrence based on meteorological drought indices in Europe, J. Hydrol., 530, 37–50, https://doi.org/10.1016/j.jhydrol.2015.09.039, 2015.

Vicente-Serrano, S. M., López-Moreno, J. I., Beguería, S., Lorenzo-Lacruz, J., Azorin-Molina, C., and Morán-Tejeda, E.: Accurate Computation of a Streamflow Drought Index, J. Hydrol. Eng., 17, 318–332, https://doi.org/10.1061/(ASCE)HE.1943-5584.0000433, 2012.

---

## Author Response (AR1)

**Dear Editor**

Please find in the table below the review comments and responses. We appreciate the constructive comments which have contributed to an improvement of our manuscript.

With kind regards, and on behalf of all co-authors,

Rhoda Odongo

**Response to reviewers**

*Propagation from meteorological to hydrological drought in the Horn of Africa using both standardised and threshold-based indices*

**General response**

We want to thank referee 1 and 2 for the critical review of our manuscript and for the positive words about our paper and its contribution. We have reflected on the comments and we have made some major revisions that have led to a significant improvement of our manuscript. We hope that the revised version provides enough detail and clarity to cover the original concerns presented. Below are our point-to-point response (in bold, with the original reviewer comment in normal format and the changes in the manuscript in italics dark blue) to the questions raised. The line changes mentioned refer to the revised manuscript unless stated otherwise.

**#Reviewer 1-RC1**

*Major comments:*

- The first thing I would suggest to the authors is that they carefully review the text to avoid several grammar errors and typographical errors prevalent in the manuscript (I list some of these errors at the end of the review as examples).

**We have carefully reviewed the manuscript text for the suggested grammatical errors and any other errors we may have missed.**

The whole manuscript has been carefully reviewed for grammar errors and typographical errors. We improved the writing style of the manuscript, in some places significantly, such as the Introduction and Methodology section. The manuscript has also gone through multiple editing and professional proofreading services. If you have any further suggestions regarding the English editing, please let us know which sentences specifically require further editing.

Please see the tracked changes document.

- Studies have developed and used a wide range of meteorological drought indices. Could authors briefly explain why they selected/prefer SPI over other indices?

**We preferred to use SPI over other indices because we learned that organizations delivering climate services to Horn of Africa like IGAD Climate Prediction and Application Centre (ICPAC) use SPI specifically for drought monitoring in its East Africa Drought Watch. Several studies (Kalisa et al., 2020; Okal et al., 2020; Dinku et al., 2007; Viste et al., 2013) in the Horn of Africa have also applied SPI. In our study, we do not only use standardised indices like SPI, we also apply threshold-based indices and compare how standardised and threshold-based indices characterise drought propagation in the region. We have clarified this in the revised version**

Added lines 205 to 209:

*'In this study, we prefer SPI over other meteorological drought monitoring indices because organizations providing climate services to the Horn of Africa, such as the IGAD Climate Prediction and Application Centre (ICPAC) use SPI specifically for drought monitoring in its East Africa Drought Watch. Several studies in the Horn of Africa have also used SPI (Kalisa et al., 2020; Okal et al., 2020; Dinku et al., 2007; Viste et al., 2013).'*

- ▪ Authors did not explain/mentioned how they identify drought events based on these standardized indices (onset and termination of drought events). Since these standardized indices encompass both droughts and non-drought periods.

**In the analysis, we tested two separate methods. The first was the standardised indices method which included standardised wet and dry periods to characterise *drought months*. As such we did not identify drought events using standardised indices, but rather the relative dryness over different accumulation periods without defining whether it's a drought event or not. The second method was the threshold-based indices, in which we applied a threshold (70th percentile) to characterise *drought events* (Heudorfer and Stahl, 2017). So here we do identify events and define these events in terms of duration. We applied a variable threshold (without pooling), so one that has a different value each month; this is similar to standardised indices that fit a distribution for each month separately. Both methods highlighted different aspects of the drought. The threshold-based duration preserved the original hydrological values and estimated how long the water shortage lasted while the standardised indices indicated relative water availability. However, both methods do say something about drought propagation, but in different ways: either as changes in duration of an event, or the accumulation period. The different ways that we defined the anomalies / droughts do not give almost similar results on drought propagation and the relation with climate and catchment characteristics. We have explicitly mentioned this in our explanation in the Methods section.**

Added lines 226 to 229:

*'All drought indices were calculated with a monthly resolution for the period 1980–2020. The standardised wet and dry periods of each indicator were included in the analysis to characterise changes in anomalies when moving through the hydrological cycle. As such, with this method, we did not define drought events, but aim to identify the anomalies over different accumulation periods.'*

Added lines 233 to 237:

*'Drought events were identified from the monthly time series of the above hydrometeorological datasets (precipitation (P), soil moisture (SM) and river discharge (Q)) using a monthly varying threshold-based approach (without pooling) i.e. an approach that has a different value for each month (this is similar to standardised indices that fit a distribution for each month separately) to account for seasonality, and defined in terms of the duration of drought.'*

- ▪ Is there any particular reason for having different interval classes in Figure 1b? If not, then I would suggest that the class interval and color scale be changed with distinct colors for the classes.

**We agree. We did this because the list of elevation values were too many and we needed to try to display all the classes, hence the use of intervals. We have changed the color scheme as suggested.**

Figure 1 has been changed with the appropriate color scheme and legend.

Please see the manuscript.

- Figure 2: The authors mentioned two drought characteristics i.e., duration and severity/deficit volume. However, the entire paper focuses exclusively on the duration of the drought. Therefore, it is recommended that the severity/deficit volume analysis be added or removed from Figure 2.

**We agree. We have removed these parts from Figure 2.**

In Figure 2 the severity/ deficit volume analysis has been removed.

Please see the manuscript.

*Minor Comments*

***We have addressed all the minor comments appropriately in the manuscript.***

- It is preferable to have different color boundaries for the countries so that the location of each country can be identified immediately.

All the figures now have different color boundaries for the three countries and the upstream catchment area.

Please see the manuscript.

- Figure 2: The resolution of the figure is quite low

All the figures resolutions has been changed to a higher resolution.

Please see the manuscript.

- I believe this paragraph should be placed in the introduction rather than on lines 124-132

The paragraph has been moved to line 36-44.

- Lines 209-211: It is appropriate to provide these results as supplementary information.

The threshold selection plots have been added to section 2.2 Supplementary Material.

- Typos and English grammar (examples)

The manuscript has been thoroughly checked for typos and English grammar issues.

Please see the tracked changes document.

- Line 15-16: "and by calculating the ratio between the threshold respectively streamflow drought duration". The use of respectively is not clear, please rephrase

Rephrased in line 13 to 18:

*'The relationship between meteorological and soil moisture is investigated by finding the SPI accumulation period that has the highest correlation between SPI and SSMI and the relationship between meteorological and hydrological drought is analysed by the SPI accumulation period that has the highest correlation between SPI and SSI time series. Additionally, we calculated these relationships with the ratio between the threshold-based meteorological drought duration and soil moisture drought duration, and the relation between threshold-based meteorological drought duration and streamflow drought duration.'*

- Line 425: We find differences in propagation from precipitation to soil moisture to also be influenced by ….

Rephrased in line 471 to 473:

*'We find that the differences in propagation from meteorological to soil moisture drought are also influenced by spatial variability in precipitation, with the wetter western part of HOA having catchments with both short and long propagation timescales, and the drier eastern having only long propagation timescales.'*

- Line 509: repetition of the sentence (As such, the dataset tends to overestimate streamflow in arid and semi-arid areas).

The repeated sentence has been removed.

Please see tracked changes document on lines 625 to 626.

**#Reviewer 2- RC2**

***Major comments***

- First of all, the question of the uncertainty relating to the data used should be addressed. Since the research work deals with modeled, re-analyzed and gridded data, coming from different temporal and spatial aggregation scales and from different models, each data has its own uncertainty and in their combined use it is not possible to verify this question, not even the different uncertainties can be compared each other. GLEAM for example uses MSWEP as input, but what can we say about GloFAS? What rainfall input does it use? If different from MSWEP, how can we compare the modeled flow data with the rainfall data of MSWEP?

**The source and quality of the data we use has indeed been a topic we discussed a lot and deserves more explicit inclusion in the manuscript. Our main objective was to work with observational data, or data that is as close to observations as possible for the entire region. For precipitation there were various good products available. We chose for MSWEP because that is consistent with GLEAM (avoiding accumulation of uncertainty resulting from different sources), which in turn estimates soil moisture using satellite imagery and a re-analysis approach. The MSWEP precipitation data was also selected due to its demonstrated good performance across Western, Eastern, and Southern Africa. It has shown a strong correlation with in-situ observations and substantial agreement with CHIRPS precipitation data (which has been popularly applied in the region because it has been found to show a good depiction of rainfall seasonality, and in a study by Musie et al., (2019), they used CHIRPS precipitation to model daily and monthly streamflow and the simulated streamflow data matched gauged streamflow observations). MWEP has better results when compared to ERA-Interim precipitation data (which was originally applied in the generation of GloFAS river discharge**

data). These findings are reported in studies by Cattani et al., (2021) and Beck et al., (2017). We chose to not use ERA5 precipitation because the quality is not good and there are no rain-gauge data assimilated into the product.

For discharge, we initially also wanted to use observation data or a data product that is as close to observations as possible (like GLEAM for soil moisture). However, the spatial and temporal coverage of observed discharge data in the region is too low for our analysis. Therefore we decided to use modelled data for discharge, but there is no dataset available that uses MSWEP for precipitation input. GloFAS uses ERA5 Land total precipitation data from EMCWF, as input to the hydrological model LISFLOOD. We could have used ERA5 Land precipitation as our precipitation data in this study (for consistency with GloFAS), but we did not want to do this because ERA5 has been found to highly underestimate/overestimate the precipitation values in the region. Fessehaye et al., (2022) tested the product for Eritrea region and found it highly underestimated precipitation values. Gleixner et al., (2020), on the other hand, tested the product against CHIRPS dataset and found it overestimated precipitation in East Africa (see a copy of their results in Figure 1 below). GloFAS is calibrated and evaluated against in-situ river discharge, but mainly for perennial rivers at mid-latitudes (Harrigan et al., 2020; Hirpa et al., 2018). When we compared the GloFAS discharge values with GRDC in-situ observations in the study region, we found that there often was a strong bias in absolute values (see Figure 2) and that the anomalies (value divided by annual mean discharge) are captured well (see below Figure 3 for two different stations in Ethiopia). As we work only with relative indices for our drought study (either standardized, or with a relative threshold), the absolute bias is not an issue in our application. Therefore, we decided to use GloFAS. We have included these figures in Supplementary Material and also included the above reasons in the manuscript.

[Figure]

*Fig. 1: Plots of ERA5 Land precipitation against CHIRPS dataset (Glexner et. al., 2020)*

[Figure]

Fig. 2: Plots of GloFAS river discharge against observations

[Figure]

[Figure]

*Fig. 3: Plots showing the discharge anomalies between the observed data and GloFAS discharge. The deviations are similar.*

Added lines 184 to 191:

*'The GloFAS dataset was selected because there is no observed river discharge data with sufficient spatial coverage and time period in the study region. Unfortunately, GloFAS uses ERA5 Land as precipitation input, which has been found to be less reliable in the HOA region than MSWEP or CHIRPS. Therefore, we tested the GloFAS dataset at the available discharge stations (with discharge values from 1981 onwards: total of 26 stations) in the HOA for bias compared to observed data. We found that while there is a bias in the absolute values, the anomalies are similar between the two datasets (see for more explanation in section 1 Supplementary Material). Since our analysis focuses on relative deviations from normal, we deemed it acceptable to use the GloFAS data to represent discharge anomalies.'*

- Another source of uncertainty is also probably due to the fact that the different standardized indices (SPI, SSMI, SSI) are calculated for each basin with reference to different probability laws. How this would have impacted the analysis?

**We indeed applied different distributions and picked the one with the best fit to calculate each of the indices. By doing this, we are not consistent between catchments, which indeed causes uncertainties when comparing values between different catchments. But, as we are only looking to compare within catchments, we prefer the better fits to the spatial consistency.**

**By the nature of the different indices, different distributions are best suited to fit the different data types. This is recognized in literature and as such we used the distributions suggested by Stagge et al., (2015) for calculation of SPI, distributions suggested by Ryu and Famiglietti, (2005) for calculation of SSMI and distributions suggested by Vicente-Serrano et al., (2012) for calculation of SSI. We have further explained this in the manuscript.**

**Like with fitting a different distribution for each variable, also fitting a different distribution for each catchment causes differences, especially in the tails (i.e. drought extremes). However, this is only relevant if one is directly comparing drought onset or severity of the same event between catchments. In our case we do not map and analyse the droughts themselves, but instead we look at the propagation in the hydrological system. To do this, we correlated the time series of the meteorological indices with several accumulation periods to the hydrological indices to find**

the best correspondence. In our opinion, in this case the benefits of having the best fit to the data for each catchment outweigh the differences in extreme droughts between catchments.

Another way to verify our results is the comparison with threshold-based indices. The results of the propagation analysis done with the threshold-based duration ratios is very similar to that based on accumulation periods of standardized indices. This indicates that the distribution fitting is expected to have a minor role.

To give the reviewer more details about the distribution fitting, we here include some extra analysis. For example looking at the fitting of precipitation data for calculation of SPI-1 for the different catchments, we can see in the figures below that the depiction of the drought months is not different. This was similar for the other indices i.e. SSMI and SSI.

[Figure]

*Fig. 4: Histogram showing the number of times each distribution was used in the calculation of SPI-1*

[Figure]

*Fig. 5: the fitting of the distribution for the calculation of SPI-1 in a catchment in (semi-)arid . The best distribution is framed in black*

The fittings for the distributions in the catchments in general were not very good, as shown for one example catchment in Figure 5, hence we decided to not force fit data for each catchment into the same distribution to avoid introducing artifacts. This is because if we do force fit a consistent distribution throughout the catchments, especially in dry catchments with lots of zero values, we anticipate worse fitting distributions and even worse results. Additionally, we are rather more interested in the catchment propagation than the spatial pattern existing among the catchments, because we would like to better identify the droughts accurately within each catchment.

See Figure 6 below for four different catchments.

[Figure]

*Fig. 6: shows the SPI-6 after fitting different distribution through the precipitation data for four catchments (locations indicated on the y-axis). The best distributions are bold*

**There was a similar representation when looking at the SSMI-1 distribution fitting. There is very minor difference in the distributions (Figure 7). Therefore, we concluded that the use of the different distributions will not affect our results. We have added a few lines about this and included these figures in the Supplementary Material.**

[Figure]

*Fig. 7: shows the SSMI-1 after fitting different distribution through the soil moisture data for four catchments(locations indicated on the y-axis) . The selected best distributions are in bold*

Added lines 211 to 216:

*'By the nature of the different indices, different distributions are best suited to fit the different data types. We used the distributions suggested by Stagge et al., (2015) for calculation of SPI, distributions suggested by Ryu and Famiglietti, (2005) for calculation of SSMI and distributions suggested by Vicente-Serrano et al., (2012) for calculation of SSI. We fitted a different distribution for each catchment, which is not a problem in our study because we analyse drought propagation with catchments and do not compare drought characteristics between catchments (see for more explanation section 2 in Supplementary Material).'*

- I understand that the research idea is the propagation of drought, but perhaps a risk analysis (see Figure 2), which also showed the quantitative evaluation and the relative characteristics of the indices, also from a spatial point of view, would have helped to sort out the problem. For example, on line 443 it is said, if I understand correctly, that the drought indices used "fail to capture the water deficit amount…" how can we know this if we do not have an illustration of the quantitative assessment of the indices?

**Line 443 was based on a literature review on the advantages and disadvantages of using standardized indices against the threshold-based indices. Which index would be better able to capture absolute water deficits was not tested in this manuscript. Looking into drought characteristics themselves and their spatial representation was beyond the scope of our analysis. It would have made the manuscript substantially longer and would have taken away the focus from the propagation results, which in our view are more interesting to study. Additionally, as we discussed in our response to the previous comment, the spatial comparison of the drought characteristics themselves has uncertainties due to the fitting of different distributions in each catchment. We have included the quantitative time series plots of the drought propagation in the supplementary material for different catchments located in different places within the region of study for the readers, but would be hesitant to draw any conclusions from these.**

Plots for the propagation analysis for three different catchments located in the humid and (semi-)arid regions of the Horn of Africa have been added to Section 3.1 in the Supplementary Material.

- In general, the quality of preparation should be improved. Some sections of the manuscript were particularly difficult to read and sometimes repetitive. For example, section 3.3 says little more than 3.3 so maybe they could be merged. In general, the paragraph on methodology should be improved. In the same way, the presentation of figures (figure 5) and tables (table 1) in points of the body of the text where the analyses have not yet been inserted are misleading.

**Thank you for this useful comment. We have carefully reviewed the manuscript and removed repetitions while also merging the suggested sections.**

Section 3.3.1 and 3.3.2 have been merged.

Methodology section writing style has been improved and reviewed.

We tried many different ways to represent the figures and table 1, and the structure presented in the manuscript was the most plausible way of doing it without having too many figures and confusing the reader.

Please see the tracked changes document.

- In general, even the figures should be improved in their quality and also in the formatting of the characters, which even when printed are too small for easy reading. I wonder why in figure 3 the same colours are always used except for panel (a) and the same for figure 6. Why don't change them or just have always the same colour? In figure 3 and 6 there are no units of measurement. Furthermore, in table 1 it is not clear to me (not even from the caption) what is the difference between the two SPI-to-SSMI (months) columns. The two columns have the same header but different values.

**In Figure 3a represents the correlation values obtained during the propagation analysis while Figures 3b-e represent propagation indices maps, hence, the different colours. Same explanation applies to Figure 6. We agree to change the figures to a similar colour scheme.**

**We have also included the units of measurements in both figures.**

**In Table 1, the different values are for each of the climate and catchment characteristics (the table was split into 4 columns to reduce the length). The values represent the mean aggregation of the selected indices of each catchment per each class of the climate and catchment**

**characteristics. We changed the structure to a vertical layout for clarity and included an explanation in the manuscript for better understanding.**

All the Figures in Figure 3 and 6 have been changed to the same color and units of measurements included in the legends.

Table 1 structure has been changed. We hope the new structure is easier to understand. The title has been changed in lines 374 to 376 to read:

*'Table 1: Mean accumulation period per catchment characteristics: mean annual upstream precipitation, upstream elevation, upstream area, geological type, and landcover type (≤4 months are considered short accumulation periods and ≥5 long accumulation periods).'*

***Minor comments***

- Line 94: what is "level 6 boundaries of HydroBASINS?

**HydroBASINS is a system that divides a large water basin into smaller sub-basins at points where two river branches meet and each has a minimum upstream area of 100 km². The sub-basins are also grouped and coded to allow for the creation of smaller sub-basins at different scales, or to move from upstream to downstream within the sub-basin network. To make this possible, HydroBASINS uses the "Pfafstetter" coding system, offering 12 nested sub-basin levels globally. We will expand the explanation of the selected level in the manuscript.**

Added lines 145 to 150:

*'HydroSHEDS is a global hydrological dataset that provides information on water drainage systems. It is based on digital elevation models (DEMs) and other geospatial data sources and is divided into several levels of detail, with level 1 being the coarsest and level 12 the finest. At each level, the dataset provides information on the location and characteristics of water bodies such as rivers, lakes and wetlands, as well as the topography of the surrounding terrain. Level 6 was chosen because it provides an average level of detail on the water drainage systems. In particular, hydrographic units (HUs) with an average size of about 10,000 square kilometres are delineated at level 6.'*

- Line 110: You probably selected the study region not because it is an "interesting" region but because, given the large number of catchments and relevant catchments attributes, it would have been suitable for the purpose of the study.

**We agree and we have added a better reason behind the selection of the study region.**

Added lines 120 to 121:

*'The diverse topography, climate seasonality and large number of catchments and relevant catchment characteristics make it a suitable region for study.'*

- Line 116: "followed by the calculation of the indices". Which indices?

**The referred indices were SPI, SSMI,SSI and drought duration indices. This has been added to the manuscript**

Added lines 127 to 130:

*'…followed by calculation of the Standardised Precipitation Index (SPI), Standardised Soil Moisture Index (SSMI), and Standardised Streamflow Index (SSI) and the threshold-based indices (Precipitation to Soil*

*moisture mean duration ratio (P/SM ratio) and Precipitation to Streamflow mean duration ratio (P/Q ratio) (Section 3.2).'*

▪ Line 146: "three hourly temporal", please clarify

**The MSWEP precipitation data has a resolution of 0.1 degrees grid with 3 hour intervals. Though for the analysis we used the daily precipitation data.**

This has been rephrased in lines 158 to 160:

*'…MSWEP has a temporal resolution of 3 hours, daily, and monthly; and a spatial resolution of 0.1 degrees. In this study we used the daily MSWEP precipitation data.'*

▪ Line 169: what "HAD" is?

**This should have been Horn of Africa (HOA).**

This has been changed to HOA

▪ Line 209: it is questionable the fact you selected 70th percentile because otherwise you have a too small number of events. The selection of the percentile deals with the severity of the drought events. Probably another message would have come from the analysis

**We agree and have rephrased the sentences.**

Added lines 243 to 247:

*'After testing different percentile values ($70^{th}$, $80^{th}$ and $90^{th}$ percentile), we selected the $70^{th}$ percentile because it could clearly capture both moderate and severe droughts. The other percentiles were eliminated because, with the high precipitation variability experienced in the region, they had too few droughts, showed a misidentification of less severe droughts, and they did not account for most of the major drought years. This made it difficult to identify patterns and trends (see section 2 Figure S9, S10 and S11 Supplementary Material).'*

▪ Line 215-218: it is not clear to me the explanation provided when P/Q is close to zero. This should happen when the duration of the meteorological drought is significantly smaller than the one for the hydrological drought, and this would happen when you have a small number of meteorological drought events or short meteorological drought events. Also I believe authors should better explain why they decided to use P/Q and P/SM ratio as indicators of drought propagation, what does this ratio means and why they did not use more convention drought propagation index (line 233 it is said that you did not consider the lag because in some previous study it is said that the largest correlation occurs at lag 0)

**We have added a more detailed description and explanation in the manuscript detailing the reasons for the use of the duration ratios and their representation.**

Added lines 274 to 284:

*'P/SM mean duration ratio represents the speed with which precipitation deficits affect soil moisture availability, and therefore, how quickly the ability of plants to access water is hampered during drought. A low ratio suggests that soil moisture is more resilient to precipitation deficits (slow soil moisture response to precipitation), which is probably related to catchment properties like soil type. A high ratio indicates that*

*precipitation deficits have a faster impact on soil moisture availability (faster soil moisture response to precipitation). We also calculated the ratio of the duration of meteorological drought and streamflow drought (P/Q) to show the propagation from meteorological to streamflow drought. P/Q mean duration ratio represents the degree to which precipitation deficits affect streamflow. A low ratio suggests that streamflow is more resilient to precipitation deficits, and meteorological droughts are buffered. A high ratio indicates that precipitation deficits have a quick response in streamflow. Also these P/Q ratios are probably influenced by catchment characteristics like subsurface storage.'*

- Line 242-243: reformulate

**We have reformulated the sentence.**

Reformulated in lines 284 to 287:

*'Overall, we favoured the use of the duration ratios to other conventional indices because these ratios can provide insight into the mechanisms through which drought propagates and the vulnerabilities of different systems to precipitation deficits (Van Loon et al., 2016). These ratios take into account the effect of precipitation deficits on soil moisture and streamflow.'*

- In my opinion figure 3 panel (b) and do not properly give the same message, much better in the case of streamflow (figure7).

**We do not understand what the reviewer meant. Please give more details so that we can make the appropriate changes.**

- Line 319: in my opinion it is improper to discuss about "precipitation variability" as you only considered the mean annual precipitation for each catchment and this is not a variability index for precipitation (also on line 425)

**We use mean annual precipitation as a descriptor of the precipitation climate as has been used in previous studies (e.g. Barker et. al., 2016) and to refer to the spatial differences in precipitation. We agree to avoid using the term "precipitation variability" where we mean spatial variability.**

This has been changed to the spatial variability in precipitation.

Lines 362 to 363:

[revised manuscript text omitted]

---

## Author Response (AR2)

**Dear Editor**

Please find below the review comments and responses. We appreciate the constructive comments which have contributed to an improvement of our manuscript.

With kind regards, and on behalf of all co-authors,

Rhoda Odongo

**Response to reviewers**

*Propagation from meteorological to hydrological drought in the Horn of Africa using both standardised and threshold-based indices*

**General response**

We want to thank referee 2 for the critical review of our manuscript and for the positive words about our paper and its contribution. We have reflected on the added comments/review and we have made some minor revisions that have led to a significant improvement of our manuscript. We hope that the revised version provides enough detail and clarity to cover the original concerns presented. The line changes mentioned refer to the revised manuscript unless stated otherwise.

**#Reviewer 2- RC2**

*Minor comments*

I would like to thank the authors for their reply to the questions posed in the first review round and for providing specific assessments to support the answers in the supplementary material.

In my opinion the research work overall is worthy of publication and that indeed it could be an example for other countries and territories affected by a severe lack of data. Precisely for this reason, I believe it is necessary to illustrate the relationship between in situ data (however few, data at the scale of the single basin must exist - see supplementary material) and those deriving from the datasets actually used in the work and the related implications. From the authors' reply I understand they are aware of this consideration, but assigning the topic to a few quick comments that refer to the supplementary material does not seem to me the best proposal. Given the level of complexity with which the article is organized, I understand the difficulty in the preparation of a specifically conceived section and I suggest, as an alternative, at least a critical discussion on the issue (maybe section 5.1?).

We agree that the best approach would be to include another section in the Discussion on the topic.

Added lines 460 to 496 under section 5.1 Data selection and limitations:

[revised manuscript text omitted]